# Hamster model for post-COVID-19 alveolar regeneration offers an opportunity to understand post-acute sequelae of SARS-CoV-2

Laura Heydemann [1,7], Małgorzata Ciurkiewicz [1,7], Georg Beythien [1], Kathrin Becker [1], Klaus Schughart [2,3], Stephanie Stanelle-Bertram[4], Berfin Schaumburg[4], Nancy Mounogou-Kouassi[4], Sebastian Beck[4], Martin Zickler[4], Mark Kühnel[5,6], Gülsah Gabriel[4], Andreas Beineke[1], Wolfgang Baumgärtner [1,7] ✉ & Federico Armando [1,7]

COVID-19 survivors often suffer from post-acute sequelae of SARS-CoV-2 infection (PASC). Current evidence suggests dysregulated alveolar regeneration as a possible explanation for respiratory PASC, which deserves further investigation in a suitable animal model. This study investigates morphological, phenotypical and transcriptomic features of alveolar regeneration in SARS-CoV-2 infected Syrian golden hamsters. We demonstrate that CK8[+] alveolar differentiation intermediate (ADI) cells occur following SARS-CoV-2-induced diffuse alveolar damage. A subset of ADI cells shows nuclear accumulation of TP53 at 6- and 14-days post infection (dpi), indicating a prolonged arrest in the ADI state. Transcriptome data show high module scores for pathways involved in cell senescence, epithelial-mesenchymal transition, and angiogenesis in cell clusters with high ADI gene expression. Moreover, we show that multipotent CK14[+] airway basal cell progenitors migrate out of terminal bronchioles, aiding alveolar regeneration. At 14 dpi, ADI cells, peri-bronchiolar proliferates, M2-macrophages, and sub-pleural fibrosis are observed, indicating incomplete alveolar restoration. The results demonstrate that the hamster model reliably phenocopies indicators of a dysregulated alveolar regeneration of COVID-19 patients. The results provide important information on a translational COVID-19 model, which is crucial for its application in future research addressing pathomechanisms of PASC and in testing of prophylactic and therapeutic approaches for this syndrome.

Severe acute respiratory syndrome coronavirus 2 (SARS-CoV-2) caused over 600 million infections and over 6.5 million fatal outcomes to this day (October 2022, WHO). Patients surviving acute COVID-19 are at risk to develop post-acute sequelae of SARS-CoV-2 (PASC)[1–3]. PASC occurs in 3-11.7% of infected individuals and is characterized by symptoms such as fatigue, headache, cognitive dysfunction, altered smell and taste, shortness of breath, and dyspnea, occurring >12 weeks after acute virus infection[4,5]. Of note, among patients with severe

---

disease requiring hospitalization, respiratory symptoms are reported with a much higher frequency. For instance, shortness of breath occurs in up to 49% and dyspnea in up to 23.3% of patients for 8–10 months after acute disease[6,7].

The pathomorphological correlates and mechanisms responsible for respiratory PASC are still not fully understood. It is known that lung fibrosis is a potential sequel to COVID-19 pneumonia[8,9]. CT scans 4 months after infection show that fibrosis develops in up to 21% of patients who survived severe COVID-19[10]. However, fibrosis cannot account for all respiratory PASC symptoms. An additional suggested mechanism is a prolonged impairment of gas-exchange capacity due to an incomplete or protracted regeneration of alveoli. SARS-CoV-2 infection of the lung causes diffuse alveolar damage (DAD), characterized by necrosis of alveolar epithelial type 1 and 2 (AT1 and AT2) cells, fibrin exudation and edema, followed by alveolar epithelial hyperplasia in later stages[11,12]. The healing of damaged alveoli and recovery of gas-exchange capacity requires the presence of progenitor cells that are able to regenerate lost AT1 cells. For a long time, it was assumed that AT1 cells are regenerated solely by proliferating and trans-differentiating AT2 cells. However, recent advances in mouse models of lung injury have shown that different airway progenitor cell types expand and mobilize to repair alveolar structures[13–16]. AT2 cells are mainly responsible for AT1 cell regeneration in homeostatic turnover and following mild injury, while airway progenitors are recruited after severe injury with marked AT1 cell loss[15,17]. The differentiation into mature AT1 cells features an intermediate step, the so-called alveolar differentiation intermediate (ADI) cell, first described to occur during AT2 to AT1 cell trans-differentiation[18–21]. ADI cells in mice are characterized by cytokeratin 8 (CK8) expression, a polygonal to elongated morphology, NFκB and TP53 activation and upregulation of genes involved in epithelial–mesenchymal transition (EMT), HIF-1α pathway, and cell cycle exit[19,22]. ADI cells have been observed in various lung injury models, e.g. bleomycin injury, neonatal hypoxia and hyperoxia, LPS injury, and Influenza A virus infection[18–20].

In homeostatic turnover and mild injury, these cells occur only transiently and differentiate into mature AT1 cells eventually, thereby restoring normal alveolar structure and function[19,21,23]. However, a pathological accumulation of ADI cells has been observed in idiopathic pulmonary fibrosis (IPF) in humans and a mouse model for progressive fibrosis, suggesting that a blockage during trans-differentiation of ADI to AT1 cells could represent a potential regenerative defect in these conditions[18,19,21,22,24]. Recently, high numbers of ADI cells have also been demonstrated in the lungs of COVID-19 patients. It has been postulated that the persistence of these cells could be responsible for unremitting hypoxemia, edema, ventilator dependence, and the fatal outcome in protracted ARDS as well as the subsequent development of fibrosis in PASC[9,22,25,26].

Since serial samples are rarely available in human observational studies, the fate of COVID-19-associated ADI cells remains elusive. Addressing this open question is of paramount importance to obtain a deeper understanding of the factors that contribute to the protracted recovery from COVID-19 facilitating the development of rational therapeutic approaches in the field of lung regenerative medicine. The development of a precise working hypothesis and subsequent pre-clinical testing of therapeutic options requires the study of sequential phases of SARS-CoV-2 infection in appropriate animal models. Among the susceptible small animal species, Syrian golden hamsters (*Mesocricetus auratus*) are well suited to study regenerative responses. They develop a distinct, but transient and non-lethal disease, in contrast to other models such as transgenic mice or ferrets[27–31]. The regeneration of lung epithelia following SARS-CoV-2 infection has not been characterized in detail in this important animal model yet.

In this work, we show further insights into the regenerative processes following SARS-CoV-2 infection. To achieve this, we characterize the proliferating epithelial cells within the lungs of infected hamsters in the acute and sub-acute phase of the infection until 14 days post infection. Our study shows that CK8+ADI cells and multipotent CK14+ airway basal cells participate in alveolar regeneration and that persistence of ADI cells at 14 dpi is associated with fibrosis in SARS-CoV-2 infected hamsters. In addition, our study provides hamster-specific marker gene lists for different alveolar cell populations, including AT1, ADI, and AT2 cells. Altogether, the results provide important information on a translational COVID-19 model, which is crucial for its application in future research addressing patho-mechanisms of PASC and for testing of prophylactic and therapeutic approaches for this syndrome.

## Results

### SARS-CoV-2-induced epithelial proliferative responses and inflammation persist beyond virus clearance

First, successful infection was confirmed by immunohistochemistry for SARS-CoV-2 nucleoprotein (NP) antigen in lung tissue. Viral antigen was found in alveolar and bronchial epithelia as well as in macrophages (Fig. 1a), as described previously[32]. Quantification of immunolabeled cells in whole lung sections peaked at 3 dpi, followed by a sharp decline at 6 dpi and virus clearance at 14 dpi (Fig. 1a). No SARS-CoV-2+ cells were detected in mock-infected animals at any time point. More information about viral titers in the lung of the animals investigated in this study can be found here[33]. Histologically, SARS-CoV-2 infected animals showed a marked, transient, broncho-interstitial pneumonia, as described previously[32,34–36]. The lesions were characterized by DAD with epithelial cell degeneration and necrosis, sloughing of alveolar cells, fibrin exudation, and heterophilic and histiocytic infiltrates. Some mock-infected animals showed small foci of mild, multifocal, interstitial inflammation composed of heterophils and macrophages, particularly at 1 dpi. The extent of inflammation in SARS-CoV-2 and mock-infected animals was quantified in total lung sections using whole slide digital image analysis of Iba-1 immunolabeling. The number of Iba-1+ cells was significantly higher in SARS-CoV-2 infected animals compared to the mock group at 1, 3, and 6, with a notable peak at 6 dpi (Fig. 1b).

The inflammatory lesions in SARS-CoV-2 infected hamsters were accompanied by a prominent epithelial proliferation (Fig. 1c). At 3 dpi, small foci of hyperplastic epithelial cells were observed within alveoli in single animals, affecting up to 1.3% of the examined lung area (Fig. 1c). At 6 dpi, large areas of prominent epithelial cell proliferation were found in all infected animals, affecting 9.3% to 39.3% of the examined lung area (Fig. 1c). Proliferating epithelial cells within the alveoli were characterized by variable morphologies, including a round cell shape typical of AT2 cells and a more polygonal to a sometimes elongated shape resembling ADI cells. Surrounding terminal bronchioles, a proliferation of cuboidal airway epithelial cells forming pods, ribbons, and tubules was observed. In the periphery, these peri-bronchiolar proliferates merged with areas of alveolar epithelial hyperplasia, showing a transition from a cuboidal to a polygonal shape (Fig. 1c). Many cells showed atypical features such as cyto- and karyomegaly, bizarrely shaped and euchromatic nuclei, as well as abundant, partly atypical, mitotic figures (Fig. 1c). At 14 dpi, multifocal areas of epithelial proliferates were still observed, affecting 2.1% to 7.2% of the examined lung area, often around terminal airways (Fig. 1c). In addition, a majority of animals (7 out of 9) showed foci of subpleural fibrosis.

In summary, SARS-CoV-2-infected hamsters showed a prominent and heterogeneous epithelial proliferative response that was still recognizable at 14 dpi, beyond virus clearance. Next, we wanted to demonstrate that alveolar AT2 cells proliferate, mobilize and differentiate into AT1 cells through the ADI cell state and that airway-derived progenitors participate in alveolar regeneration, possibly through a transitional AT2 or ADI cell state, in the Syrian golden hamster.

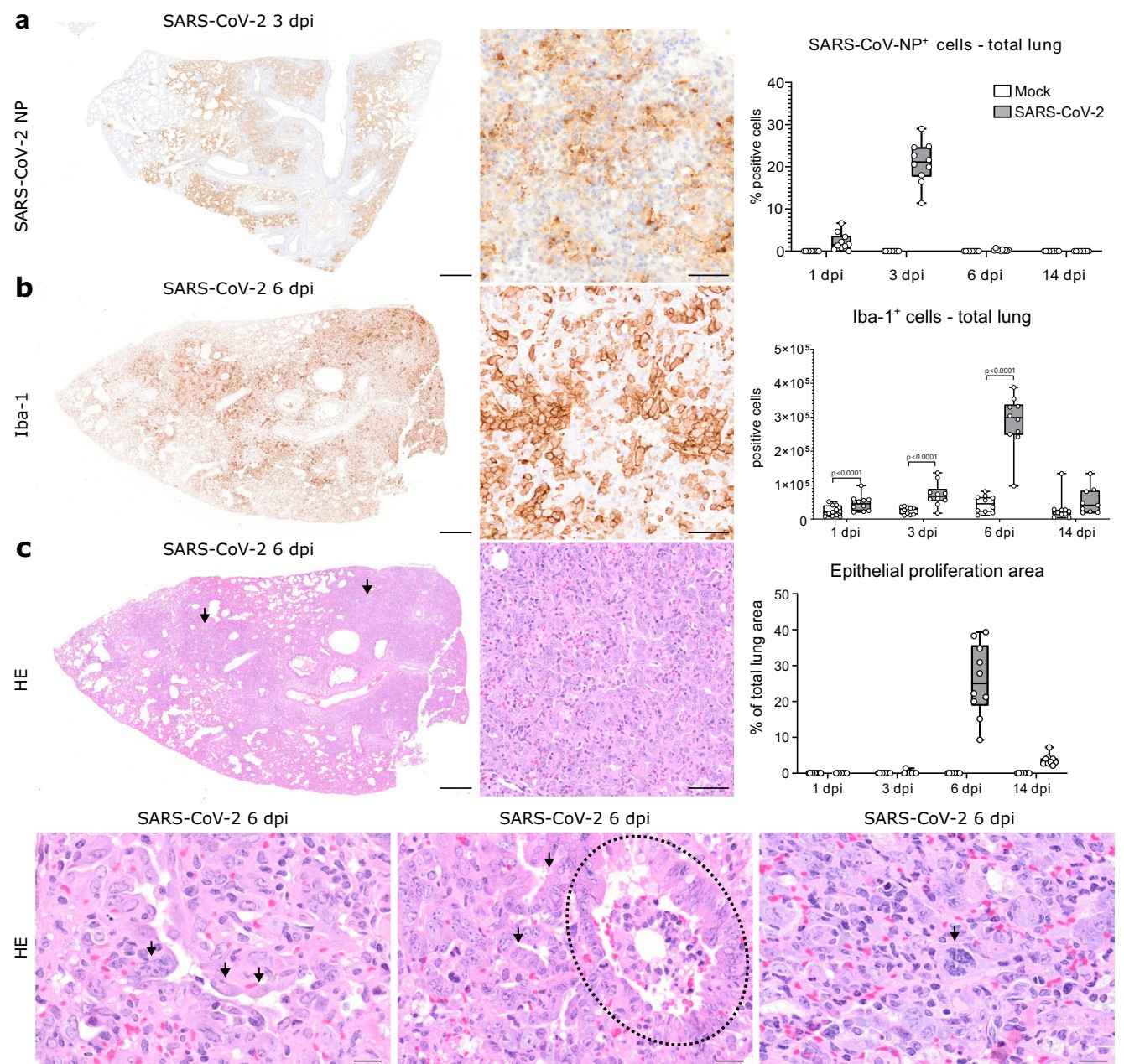

**Fig. 1 | SARS-CoV-2 infection causes a marked epithelial proliferative response in the hamster lung. a** Representative images showing SARS-CoV-2 nucleoprotein (NP) immunolabeling in one right lung lobe of an infected hamster at 3 days post infection (dpi). Left panel: overview. Central panel: higher magnification of viral antigen (brown signal) in the alveoli. Right panel: quantification of SARS-CoV-2 NP+ cells. **b** Representative images showing ionized calcium-binding adapter molecule 1 (Iba-1) immunolabeling in one right lung lobe of an infected hamster at 6 dpi. Left panel: overview. Central panel: higher magnification of macrophages/histiocytic cells (brown signal) in the affected alveoli. Right panel: Quantification of Iba-1+ macrophages/histiocytes. **c** Representative images showing histopathological lesions in a lung lobe of a SARS-CoV-2 infected hamster at 6 dpi on hematoxylin and eosin (HE) stained sections. Top left panel: overview of one right lung lobe displaying large areas of alveolar consolidation (arrows). Top central panel: higher magnification showing prominent epithelial proliferation. Top right panel: quantification of epithelial proliferation (percentage of affected area relative to total lung area). Bottom left panel: strings of plump polygonal or elongated cells lining alveolar septa (arrows). Bottom central panel: the proliferation of cuboidal airway epithelial cells forming ribbons and tubules (arrows) surrounding terminal bronchiole (dotted line). Bottom right panel: within alveolar proliferation foci, there are cells displaying karyomegaly and atypical mitotic figures (arrow). Data are shown as box and whisker plots. The bounds of the box plot indicate the 25th and 75th percentiles, the bar indicates medians, the whiskers indicate minima and maxima, dots indicate individual values. Data from Iba-1 quantification was tested by two-tailed Mann–Whitney $U$ test. A $p$ value of ≤0.05 was considered significant. $N = 10$ animals/group for mock and SARS-CoV-2 respectively. Source data are provided as a Source Data file. Scale bars: 500 µm (overviews in **a**–**c**), 50 µm (high magnifications in **a**–**c**), 20 µm (high magnifications in lower panel in **c**).

## CK8+ ADI cells frequently express TP53 and persist until 14 dpi following SARS-CoV-2-induced DAD in hamsters

ADI cells are reported to originate from AT2 and/or a particular subset of club cells expressing MHCII[19]. The AT2 to ADI cell trans-differentiation process is characterized by gradual down-regulation of AT2 cell markers, expression of CK8 and cell cycle exit markers, as well as a morphologic transition from a round to a polygonal to elongated shape[19,20]. In the following, we focused on the first part of this AT2-ADI-AT1 trajectory (Fig. 2a).

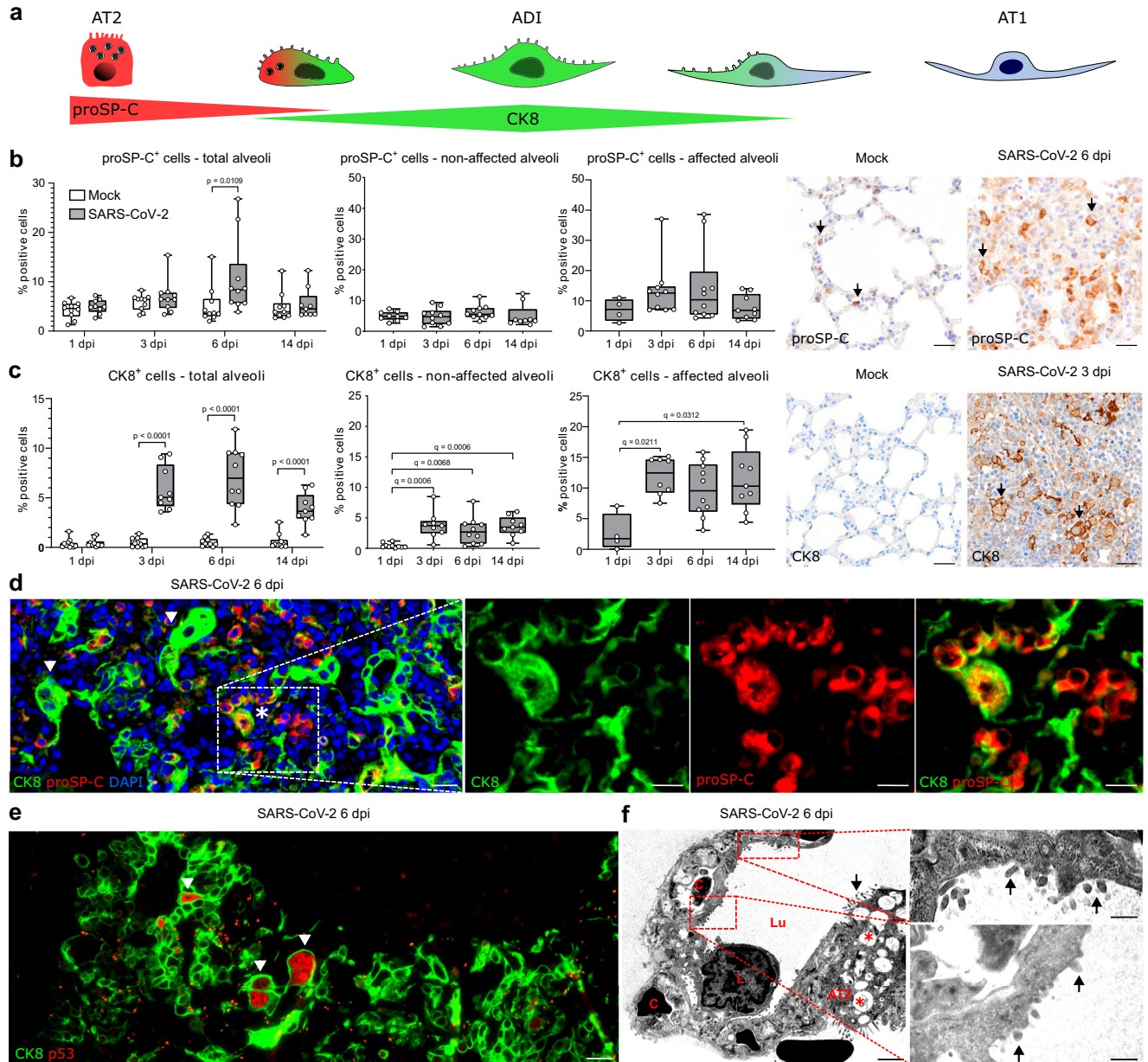

**Fig. 2 | Alveolar differentiation intermediate (ADI) cells in SARS-CoV-2 infected hamsters. a** Schematic illustration of the trans-differentiation process from alveolar pneumocytes type 2 (AT2) to alveolar pneumocytes type 1 (AT1) through the alveolar differentiation intermediate (ADI) state. **b, c** Quantification of pro-surfactant protein C (proSP-C)⁺ AT2 cells (**b**) and cytokeratin (CK) 8⁺ ADI cells (**c**) within total alveoli, non-affected alveoli, and affected alveoli as well as pictures of immunolabelling (brown signal, arrows) in the alveoli of mock and SARS-CoV-2 infected hamsters. **d** Double immunofluorescence of an alveolar proliferation focus in a SARS-CoV-2 infected hamster at 6 days post infection (dpi). Cells are labeled with CK8 (green) and proSP-C (red). There are numerous proSP-C⁻CK8⁺ADI cells, some showing hypertrophy and elongated cytoplasmic processes (arrowheads) and single, round proSP-C⁺CK8⁺ cells (asterisk). **e** Double immunofluorescence of an alveolar proliferation focus in a SARS-CoV-2 infected hamster at 6 dpi. Cells are labeled with CK8 (green) and cell cycle exit marker TP53 (red). The arrowheads show polygonal, large, bizarre TP53⁺ ADI cells. **f** Transmission electron microscopy

(TEM): alveoli of a SARS-CoV-2 infected hamster at 6 dpi. A basement membrane separates AT1 cells from endothelial cells lining capillary spaces (C) containing erythrocytes. A leukocyte (L), as well as an AT2 cell with apical microvilli (arrow) and numerous intracytoplasmic multi-lamellar bodies (red asterisks), are shown. Red boxes and high magnification show cells with flattened and elongated morphology of AT1 cells, with characteristics of AT2 cells, such as microvilli (arrows). Quantification data are shown as box and whisker plots. The bounds of the box plot indicate the 25th and 75th percentiles, the bar indicates medians, the whiskers indicate minima and maxima, dots indicate individual values. Statistical analysis was performed by two-tailed Mann–Whitney $U$ test. For multiple comparisons between time points, a Benjamini–Hochberg correction was applied. $P$- and $q$ values $\leq 0.05$ were considered significant. $N = 10$ animals/group for mock and SARS-CoV-2 respectively. Source data are provided as a Source Data file. Scale bars: 25 μm (**b**, **c**), 20 μm (overview in **d**, **e**), 10 μm (high magnification in **d**), 2000 nm (low magnification in **f**), 500 nm (high magnification in **f**).

First, we detected proSP-C⁺ AT2 and CK8⁺ ADI cells using immunohistochemistry. Quantification was performed within total alveoli first, followed by a separate analysis in areas showing inflammation and/or epithelial proliferation (termed "affected alveoli") and histologically unremarkable alveoli (termed "non-affected alveoli"). proSP-C

expression was detected in cells with a round shape lining alveolar septa. In mock-infected animals, the number of proSP-C⁺ cells was constant at all investigated time points (Fig. 2b). In SARS-CoV-2 infected animals, the total number of proSP-C⁺ cells increased significantly at 6 dpi, which was caused by an increase within affected

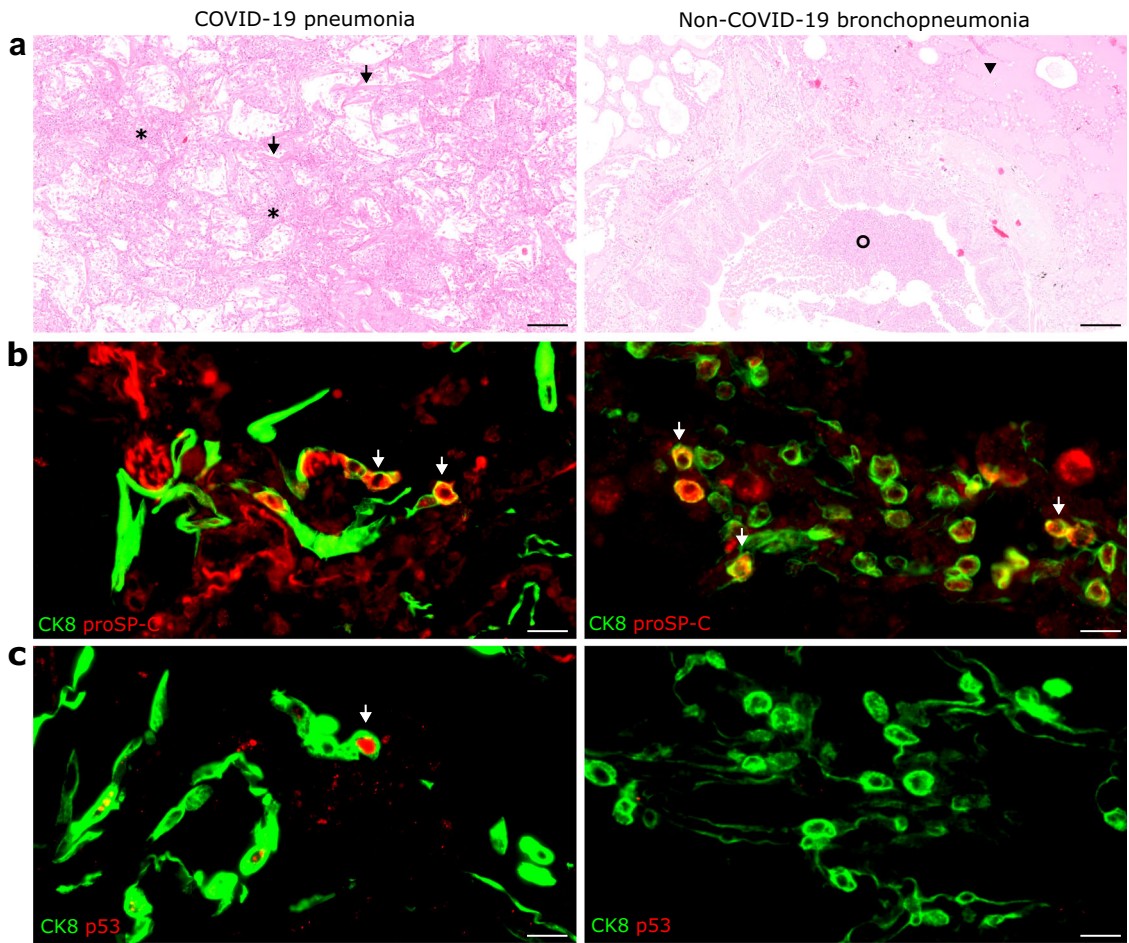

**Fig. 3 | Alveolar differentiation intermediate (ADI) cells in COVID-19 and non-COVID-19 pneumonia. a** Representative images showing histopathological lesions in a COVID-19 patient (left) and in non-COVID-19 bronchopneumonia case (right). COVID-19 is characterized by diffuse alveolar damage (DAD) with hyaline membranes (arrows) and alveolar spaces filled with sloughed epithelial cells, leukocytes, and edema (asterisks). Non-COVID-19 bronchopneumonia was characterized by intraluminal suppurative exudate (circle) and alveolar edema (arrowhead) without DAD. **b** Representative image of double immunofluorescence for the ADI marker cytokeratin 8 (CK8, green) and the AT2 marker pro-surfactant protein C (proSP-C, red) in a COVID-19 (left) and non-COVID-19 bronchopneumonia (right) sample. Cells with a round morphology express both markers (arrows). **c** Representative image of double immunofluorescence for the ADI marker CK8 (green) and the cell cycle exit marker TP53 (red) in a COVID-19 (left) and a non-COVID-19 bronchopneumonia (right) sample. ADI cells in COVID-19 patients express TP53 (arrow), while ADI cells in the non-COVID-19 bronchopneumonia case are negative. Scale bars: 200 μm and 20 μm (**b**, **c**).

alveoli. proSP-C⁺ cells were found in small groups within inflammatory foci (Fig. 2b). Scattered proSP-C⁺ cells were observed in close proximity to terminal bronchioles. Interestingly, the majority of cells within the epithelial proliferates at 6 dpi were proSP-C⁻.

CK8 was ubiquitously expressed in the apical cytoplasm of luminal cells within bronchi, bronchioles, and terminal bronchioles in all animals. In the alveoli of mock-infected animals, rare elongated CK8⁺ cells were observed, making up <1% of total alveolar cells. In SARS-CoV-2 infected animals however, CK8 was abundantly expressed within the epithelial proliferative foci at 3, 6, and 14 dpi and the number of CK8⁺ cells in total alveoli was significantly increased compared to the mock group (Fig. 2c). Importantly, increased numbers of CK8⁺ cells were detected within affected and non-affected alveoli. Within affected alveoli, the relative numbers of CK8⁺ cells remained constantly elevated throughout the investigation period. CK8⁺ cells displayed a variable cell morphology including round, polygonal, as well as elongated shapes with thin cytoplasmic processes (Fig. 2c).

Double-labeling for proSP-C and CK8 demonstrated AT2 to ADI cell transition. At 3 dpi, numerous proSP-C⁺CK8⁻ cells and rare proSP-C⁺CK8⁺ cells with a round AT2 cell morphology were observed within affected alveoli (Supplementary Fig. 1A), whereas proSP-C⁻CK8⁺ elongated cells were very rare. At 6 dpi, affected alveoli contained occasional proSP-C⁺CK8⁻ and proSP-C⁺CK8⁺ round cells (Fig. 2d). These cells were intermingled with high numbers of proSP-C⁻CK8⁺ cells, which showed various morphologies ranging from round AT2-type to polygonal ADI-type cells as well as bizarre, irregularly shaped cells with karyomegaly. Moreover, elongated proSP-C⁻CK8⁺ cells with AT1-type morphology were occasionally observed (Supplementary Fig. 1B). At 14 dpi, numerous proSP-C⁺CK8⁺ cells with AT2 morphology as well as occasional proSP-C⁻CK8⁺ polygonal cells were still detected in alveoli, including morphologically non-affected alveoli (Supplementary Fig. 1C).

Once AT2 cells enter the ADI state, they exit the cell cycle to allow AT1 trans-differentiation[21,22]. At 6 and 14 dpi, CK8⁺ cells in SARS-CoV-2 infected animals expressed nuclear TP53, indicative of cell cycle arrest and DNA repair (Fig. 2e) (Supplementary Fig. 2A, B). TP53 expression was particularly frequent in polygonal, large, bizarre, and occasionally bi-nucleated cells (Supplementary Fig. 2A, B). Of note, no TP53 expression was observed in the rare CK8⁺ cells in the alveoli of mock-infected animals. Our findings demonstrated that the transition between AT2 and ADI cells in SARS-CoV-2 infected hamsters features: (i) transient co-expression of proSP-C and CK8, (ii) changes in cell morphology from round to elongated as well as (iii) expression of cell cycle arrest markers.

In functional regeneration, ADI cells transdifferentiate into mature AT1, which assume an elongated morphology with thin cytoplasmic processes required for adequate gas-exchange. In the following, we focused on the last part of this AT2-ADI-AT1 trajectory. Double-labeling with AT1 cell markers described for human and mouse (AGER, AQP5, PDPN) was not possible since the tested antibodies failed to specifically label AT1 cells in the hamster. For this reason, we performed transmission electron microscopy to demonstrate ADI to AT1 cell transition. In normal conditions, AT1 cells are characterized by a flattened morphology with slender processes containing a moderately electron-dense, organelle-poor cytoplasm and a round to oval nucleus with a moderate amount of peripheral heterochromatin[37,38]. AT2 cells are characterized by a round morphology, an apico-basal polarity, and a moderately electron-dense cytoplasm rich in the rough endoplasmic reticulum and free ribosomes. In addition, AT2 cells possess apical microvilli as well as membrane-bound vesicles containing multiple concentric membrane layers (multi-lamellar bodies)[37,38]. In SARS-CoV-2 infected animals, proliferative foci at 6 dpi contained numerous AT2 cells (Supplementary Fig. 3A) as well as numerous hypertrophic epithelial cells with a variable cell morphology resembling ADI cells (Supplementary Figs. 3B, 4A, B). Most importantly, cells sharing AT1 and AT2 cell features were observed in the alveolar lining at the edges of proliferative foci. The cells showed the flattened and elongated morphology of AT1 cells, but also characteristics of AT2 cells, such as microvilli on the cell surface (Fig. 2f, Supplementary fig. 3c, d). Similar ultrastructural findings were present in COVID-19 patients[39]. The present findings demonstrate that the last part of the AT2-ADI-AT1 trajectory also occurs in SARS-CoV-2-infected hamsters.

Finally, we sought to confirm that the ADI cells detected in the hamster share features with ADI cells in COVID-19 patients. For this, we used lung samples obtained from three patients with lethal COVID-19 ARDS. In addition, a fourth lung sample obtained from a lobectomy of a non-COVID case was used. Histologically, the lungs from all lethal COVID-19 ARDS cases showed features of moderate to severe, acute DAD, characterized by necrosis and sloughing of alveolar cells, fibrin exudation, hyaline membranes, alveolar edema and mild to moderate neutrophilic infiltrates (Fig. 3a). In the non-COVID-19 sample, suppurative bronchopneumonia was diagnosed, characterized by neutrophilic and histiocytic infiltrates in bronchioles and alveolar lumina (Fig. 3a). Immunolabeling showed the presence of round proSP-C⁺CK8⁺ cells and polygonal to elongated proSP-C⁻CK8⁺ cells, representing the different stages of ADI cells, in all lethal COVID-19 ARDS samples as well as the non-COVID-19 bronchopneumonia sample (Fig. 3b). Interestingly, CK8⁺ ADI cells expressing TP53 have only detected the three lethal COVID-19 ARDS samples, while no TP53 co-expression was detected in the ADI cells of the non-COVID-19 case (Fig. 3c).

In conclusion, ADI cells are a feature of alveolar regeneration following SARS-CoV-2-induced DAD in lethal COVID-19 and its Syrian golden hamster model. These cells were also detected in low numbers under non-infectious conditions in the hamster and in a human sample with suppurative bronchopneumonia, confirming that ADI cells participate in physiological turnover and alveolar repair regardless of the etiology in both species. Importantly, only ADI cells from SARS-CoV-2 infected hamsters and humans expressed TP53, hinting at a prolonged block of these cells in the intermediate state.

## Multipotent airway-derived CK14⁺ progenitors contribute to alveolar regeneration following SARS-CoV-2-induced DAD in hamsters

It is well accepted that upon severe alveolar injury, both AT1 and AT2 cells can be replenished by airway progenitors (Fig. 4a)[15,17,40–42]. In the next step, we characterized the contribution of airway progenitors to alveolar regeneration in SARS-CoV-2-infected hamsters. As described earlier, histopathological lesions at 6 and 14 dpi included foci of prominent alveolar epithelial proliferation with airway-like morphology that were frequently in anatomic continuity with bronchiolar-alveolar junctions. Thus, we determined (1) the cellular origin of these proliferates and (2) whether these progenitors differentiate into AT2 or ADI cells after migrating into the alveoli.

Multiple airway progenitor cell types have been reported to contribute to alveolar regeneration, including proSP-C⁺SCGB1A1⁺ broncho-alveolar stem cells (BASCs), ΔNP63⁺CK5⁺ distal alveolar stem cells (DASCs), ΔNP63⁺CK5⁺CK14⁺ basal cells, and SCGB1A1⁺ club cells[42–44]. First, our aim was to identify these cell types in the distal airways of hamsters. The predominant basal cell type was CK14⁺, followed by CK14⁺ΔNP63⁺ cells (Supplementary Fig 5). ΔNP63⁺CK5⁺CK14⁺ cells were rare in the distal airways (Supplementary Fig 5). We did not detect ΔNP63⁺CK5⁺ DASCs, CK5⁺ cells, or SPC⁺SCGB1A1⁺ BASCs in the distal airways of hamsters. In addition to basal cell types, SCGB1A1⁺ club cells were detected in high numbers in distal airways.

In the peri-bronchiolar proliferation foci of SARS-CoV-2 infected animals at 6 dpi, the majority of cells were CK14⁺, while CK14⁺ΔNP63⁺ cells were rare (Supplementary Fig. 6). CK5⁺, CK5⁺ΔNP63⁺ or CK14⁺CK5⁺ cells were not detected within these areas (Supplementary Fig. 6). SCGB1A1 expression was absent in the peri-bronchiolar proliferates at 6 dpi, but abundantly present at 14 dpi. Therefore, we focused our further quantitative analysis on CK14⁺ airway basal cells and SCGB1A1⁺ club cells.

In mock-infected hamsters, the number of CK14⁺ cells in the airways remained unchanged over the observation period (Fig. 4b). SARS-CoV-2 infection caused a marked proliferation of CK14⁺ cells in the airways, which peaked at 6 dpi and remained elevated until 14 dpi. The number of CK14⁺ cells in total alveoli was significantly increased compared to the mock group at 3, 6, and 14 dpi, mirroring the increase in the airways (Fig. 4b). CK14 was expressed by the majority of cells in the peri-bronchiolar proliferation forming pods and tubules continuous with terminal bronchioles at 6 dpi. At 14 dpi, the peri-bronchiolar proliferates were only partly CK14⁺ (Fig. 4b).

In contrast to the CK14⁺ progenitors, we observed no major contribution of club cells in the alveolar proliferative response during early infection (Fig. 4c). The number of SCGBA1⁺ club cells in the airways remained similar in mock-infected animals at all time-points. In SARS-CoV-2 infected animals, the number of SCGB1A1⁺ cells in the airways was mildly increased compared to mock at 6 dpi (Fig. 4c). SCGB1A1 was not expressed in alveolar proliferation areas at 3 and 6 dpi. Interestingly, SCGB1A1⁺ cells significantly increased in the alveoli of SARS-CoV-2 infected animals at 14 dpi. The expression was limited to the airway-like, peri-bronchiolar proliferation areas, in which up to 40% of cells were SCGB1A1⁺ club cells (Fig. 4c).

Therefore, it was concluded that CK14⁺ cells are the airway progenitors that mainly contribute to alveolar regeneration in SARS-CoV-2-infected hamsters. These cells probably have their origin in a common ΔNP63⁺CK5⁺CK14⁺ basal cell pool, but represent a subset that loses CK5 and partly ΔNP63 expression upon migration into the alveoli.

Next, we determined the fate of the CK14⁺ cells in the alveoli. Double-labeling with proSP-C revealed clusters of CK14⁺proSP-C⁺ cells in the peri-bronchiolar pods and occasionally within the lining of terminal bronchioles. This indicates a potential differentiation of airway progenitors towards the AT2 lineage (Fig. 4d; Supplementary Fig. 7A, B). At the edges of the peri-bronchiolar proliferates, some CK14⁺ cells showed a transition from a cuboidal to an elongated shape typical of ADI cells. Co-staining with CK8 showed a gradual phenotypical change in the direction of alveoli. Cells exiting the bronchiole showed a cuboidal morphology and a diffuse cytoplasmic CK14 expression. Towards alveoli, the cuboidal cells co-expressed CK14 and CK8. More distally, cells became more elongated and were characterized by CK14⁻CK8⁺ immunolabeling (Fig. 4e; Supplementary Fig. 7C, D). Therefore, we concluded that airway progenitors can differentiate into AT2 but also directly into the ADI state. These transitions were mainly

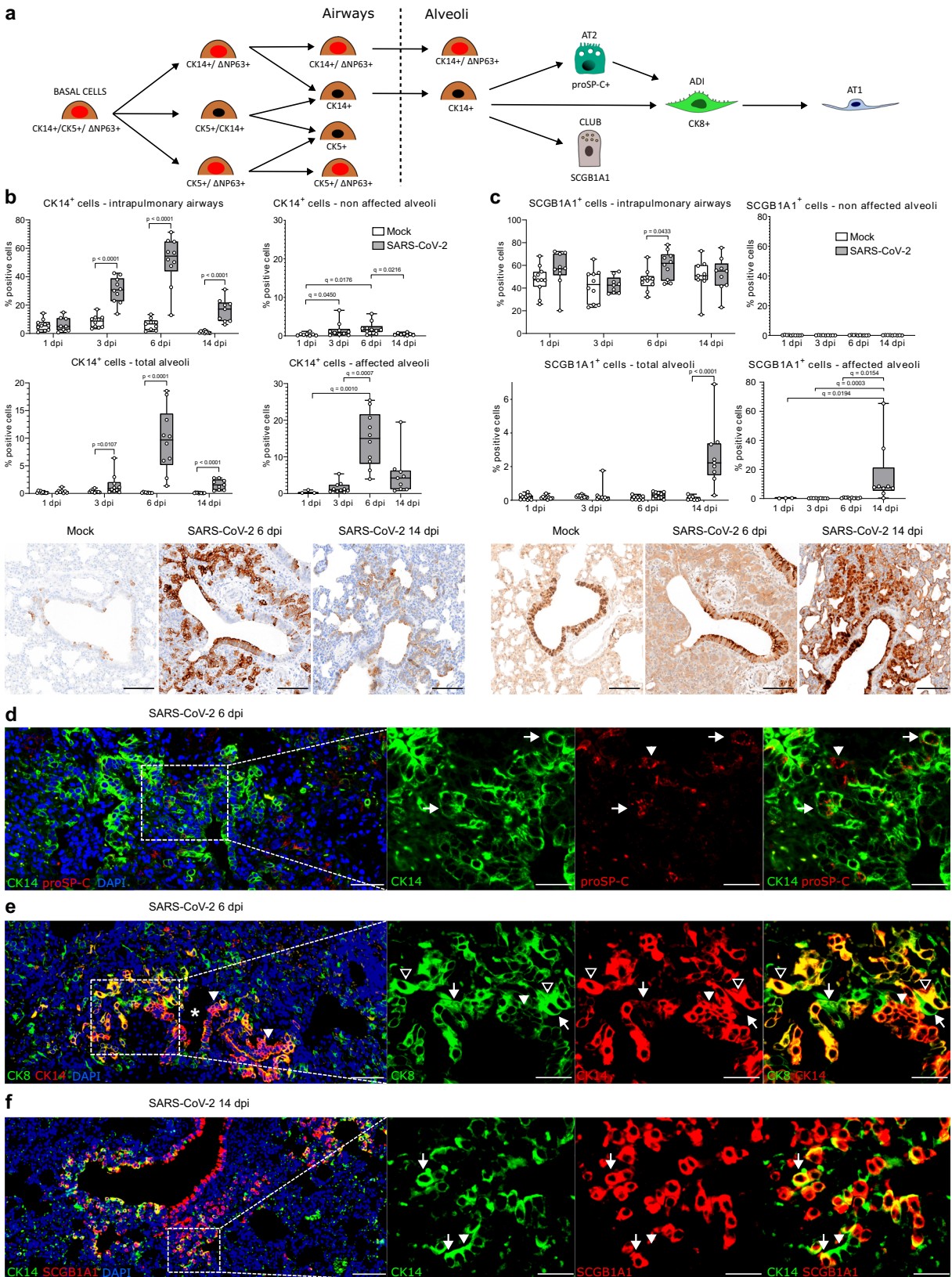

observed at 6 dpi. In contrast, at 14 dpi, peri-bronchiolar CK14+ cells partly co-expressed SCGB1A1, indicating a club cell differentiation (Fig. 4f; Supplementary Fig. 7E, F). Hence, we concluded that the increased number of alveolar SCGB1A1+ cells we observed at this time point was most likely the result of in situ differentiation of CK14+ cells. However, we cannot exclude that SCGB1A1+ club cells also proliferated

and migrated out of the bronchioles to give rise to alveolar cells at 14 dpi.

In summary, our findings indicate that multipotent CK14+ airway basal cell progenitors, probably arising from a CK14+CK5+ΔNP63+ basal cell pool, proliferate and migrate to alveoli following SARS-CoV-2 induced DAD in hamsters. These cells have the potential to

**Fig. 4 | Airway basal cells participate in alveolar regeneration in SARS-CoV-2 infected hamsters. a** Proposed trajectory of airway basal cells towards alveolar cells. Upon severe alveolar damage, rare ΔNp63⁺CK14⁺ and frequent CK14⁺ basal cells differentiate into alveolar pneumocytes type 2 (AT2) and/or alveolar differentiation intermediate (ADI) cells, particularly at 6 dpi. At 14 dpi, CK14⁺ basal cells differentiate into secretoglobin 1A1⁺ (SCGB1A1) club cells within peribronchiolar alveolar proliferates. **b, c** Quantification of CK14⁺ basal cells (**b**) and SCGB1A1⁺ club cells (**c**) within intrapulmonary airways, total alveoli, non-affected alveoli, and affected alveoli. Pictures of immunolabeled cells (brown signal) in the bronchioles and peribronchiolar proliferates in mock and SARS-CoV-2 infected hamsters at 6 and 14 days post infection (dpi), taken from the same location for both stains. **d** Double immunofluorescence for CK14 (green) and pro-surfactant protein C (proSP-C, red) in a peribronchiolar proliferation area in a SARS-CoV-2 infected hamster at 6 dpi. Arrowhead: proSP-C⁺ AT2 cell. Arrows: double-labeled airway progenitors differentiating into proSP-C⁺ AT2 cells. **e** Double immunofluorescence

for CK14 (red) and CK8 (green) in a peribronchiolar proliferation area in a SARS-CoV-2 infected hamster at 6 dpi. The transition from CK14⁺ airway basal cells forming a pod (white arrowhead) to CK14⁺CK8⁺ cells differentiating into elongated ADI cells (open arrowheads) and CK14⁻CK8⁺, elongated ADI cells (arrows). **f** Double immunofluorescence for CK14 (green) and SCGB1A1 (red) in a peribronchiolar proliferation area in a SARS-CoV-2 infected hamster at 14 dpi. Transition from CK14⁺ airway basal cells (arrowhead) to CK14⁺SCGB1A1⁺ club cells (arrows) is shown. Quantification data are shown as box and whisker plots. The bounds of the box plot indicate the 25th and 75th percentiles, the bar indicates medians, and the whiskers indicate minima and maxima. Dots indicate individual values. Statistical analysis was performed by two-tailed Mann−Whitney *U* test. For multiple comparisons between time points, a Benjamini–Hochberg correction was applied. *P*- and *q*-values ≤ 0.05 were considered significant. *N* = 10 animals/group for mock and SARS-CoV-2 respectively. Source data are provided as a Source Data file. Scale bars: 100 μm (**c**, **d**), 50 μm (overview in **d**−**f**), 25 μm (high magnification in **d**−**f**).

---

differentiate into distinct lineages, including AT2, ADI, and club cells, depending on the timing and localization.

## Hamsters show dysregulated alveolar regeneration and fibrosis following SARS-CoV-2-induced DAD

SARS-CoV-2 NP antigen was no longer detectable in the lung at 6 dpi. However, ADI cells and airway progenitors were still present in the alveoli at 14 dpi, indicating an ongoing regeneration process with incomplete restoration of alveolar structures at this time-point. Moreover, 7 out of 9 animals showed multifocal, sub-pleural, variably sized, well-demarcated areas with aggregates of spindle cells and abundant, pale, fibrillary, extracellular material (Fig. 5a). Azan staining confirmed the deposition of collagen in these areas (Fig. 5b). Immunohistochemistry for α-smooth muscle actin (α-SMA) demonstrated the presence of myofibroblasts (Fig. 5c). The fibrotic areas encompassed from 0.59 to 2.35% of the evaluated lung tissue area (Fig. 5e). Next, we wanted to determine if ADI cells and fibrosis at 14 dpi are locally associated with M2-polarized macrophages in hamsters. CD204⁺ M2-type macrophages were frequently detected within and around fibrotic areas (Fig. 5d). The number of CD204⁺ cells was significantly higher in SARS-CoV-2 infected animals compared to the mock group at 3, 6, and 14 dpi (Fig. 5f).

In summary, the findings revealed an incomplete restoration of alveolar structures with ADI cells and M2-type macrophages, as well as sub-pleural fibrosis, still detectable two weeks after infection.

## Single-cell transcriptome analysis supports ADI cell persistence following SARS-CoV-2-induced DAD in hamsters

As described above, we demonstrated that ADI cells with features previously described in mouse models of lung regeneration as well as in COVID-19 patients are participating in alveolar regeneration following SARS-CoV-2 infection of hamsters. To support this observation we wanted to detect ADI also on a transcriptome level, using data from an independent experiment. In order to do so, we re-analyzed a previously published single-cell RNASeq gene expression dataset (GSE162208) from lungs of SARS-CoV-2-infected Syrian golden hamsters[45]. The experiment was performed with a study design similar to this investigation. We focused our analysis on data from SARS-CoV-2-infected animals sacrificed at 5 and 14 dpi. First, we generated a Uniform Manifold Approximation and Projection (UMAP) embedding using the Seurat package (functions: RunPCA, FindNeighbors, FindClusters[46–51]) for clustering to visualize and identify cell populations based on similar and known/presumed expression profiles. We then identified alveolar cells based on the expression of AT1 and AT2 markers (*Rtkn2* and *Lamp3*, respectively), as described in the original publication[45] (Fig. 6a–f). These cells were then re-clustered resulting in 7 and 11 clusters at 5 and 14 dpi, respectively (Fig. 6b–g).

At 5 dpi, significantly high mean expression levels of *Rtkn2* were seen in clusters 2 and 6 (Fig. 6c). Significantly high mean expression

levels of *Lamp3* were detected in clusters 0, 3, and 5 (Fig. 6d). Clusters 1 and 4 did not show significantly high mean expression levels for any of the two markers. Based on this, clusters 2 and 6 were considered AT1, clusters 0, 3, and 5 AT2, and clusters 1 and 4 unknown. Interestingly, clusters 2, 5 and 6 showed a variably high double expression of *Rtkn2* and *Lamp3* (Fig. 6e).

At 14 dpi, significantly high mean expression levels of *Rtkn2* were seen in clusters 2, 5, and 7 (Fig. 6h). Significantly high mean expression levels of *Lamp3* were detected in clusters 4, 6, and 8 (Fig. 6i). Clusters 0, 1, 3, 9, and 10 did not show a significantly high mean expression level for neither *Rtkn2* nor *Lamp3*. Based on this, clusters 2, 5, and 7 were considered AT1, clusters 4, 6, and 8 AT2, and clusters 0, 1, 3, 9, and 10 unknown. As for 5 dpi, a variably high double expression of *Rtkn2* and *Lamp3* was detected in multiple clusters (0, 1, 2, 4, 5, 7, 9) as shown in Fig. 6j. Cluster 10 did not show any expression of the investigated genes.

As mentioned before, ADI cells are known to arise from AT2, enter this transient distinct ADI cell state and then transdifferentiate into AT1. For this reason "early" ADI cells might show an overlap in gene and protein expression with AT2 cells, as demonstrated by immunofluorescence double labeling in this work. It has been reported that the full ADI cell state is distinct and ADI cells could form a separate cluster independent from AT2 or AT1 clusters[19]. "Late" ADI cells might show an overlapping gene and protein expression with AT1 cells. Based on this, our hypothesis was that different stages of ADI cells could be located in clusters of the scRNA-seq analysis showing a double expression of AT1 and AT2 markers or in those clusters that were not clearly identifiable as AT2 or AT1.

Therefore, we analyzed gene expression profiles of all clusters in order to determine the distribution of cells expressing ADI genes. First, we identified the top 10 differentially expressed genes (DEGs) in each cluster (Supplementary Data 1 and 2) and compared the sets of DEGs with gene signatures described in mouse models of lung regeneration[19,20] as well as COVID-19 patients[9]. Within the DEGs, we detected genes typically expressed by AT1, AT2, ADI cells, club cells, or ciliated cells in mice and/or humans. We then generated a list of candidate marker genes for these cell types in the hamster. Next, we evaluated the expression of these candidate markers within the clusters and removed genes with low specificity from the lists. The final, hamster-specific marker gene lists are given in Supplementary Table 1 and the details of the expression distribution within clusters are shown as feature plots and ridge plots in Supplementary Figs. 8–27. We then applied the module score algorithm (function AddModuleScore from Seurat package[46–51]) with these sets of marker genes on the clusters detected at 5 and 14 dpi.

At 5 dpi, high module scores for AT1 cell markers were seen in cluster 6 and partly in cluster 2, in line with the distribution of *Rtkn2* expression (Fig. 7a). High module scores for AT2 cell markers were found in clusters 0, 1, 3, 4 and 5 (Fig. 7b), partially in line with *Lamp3*

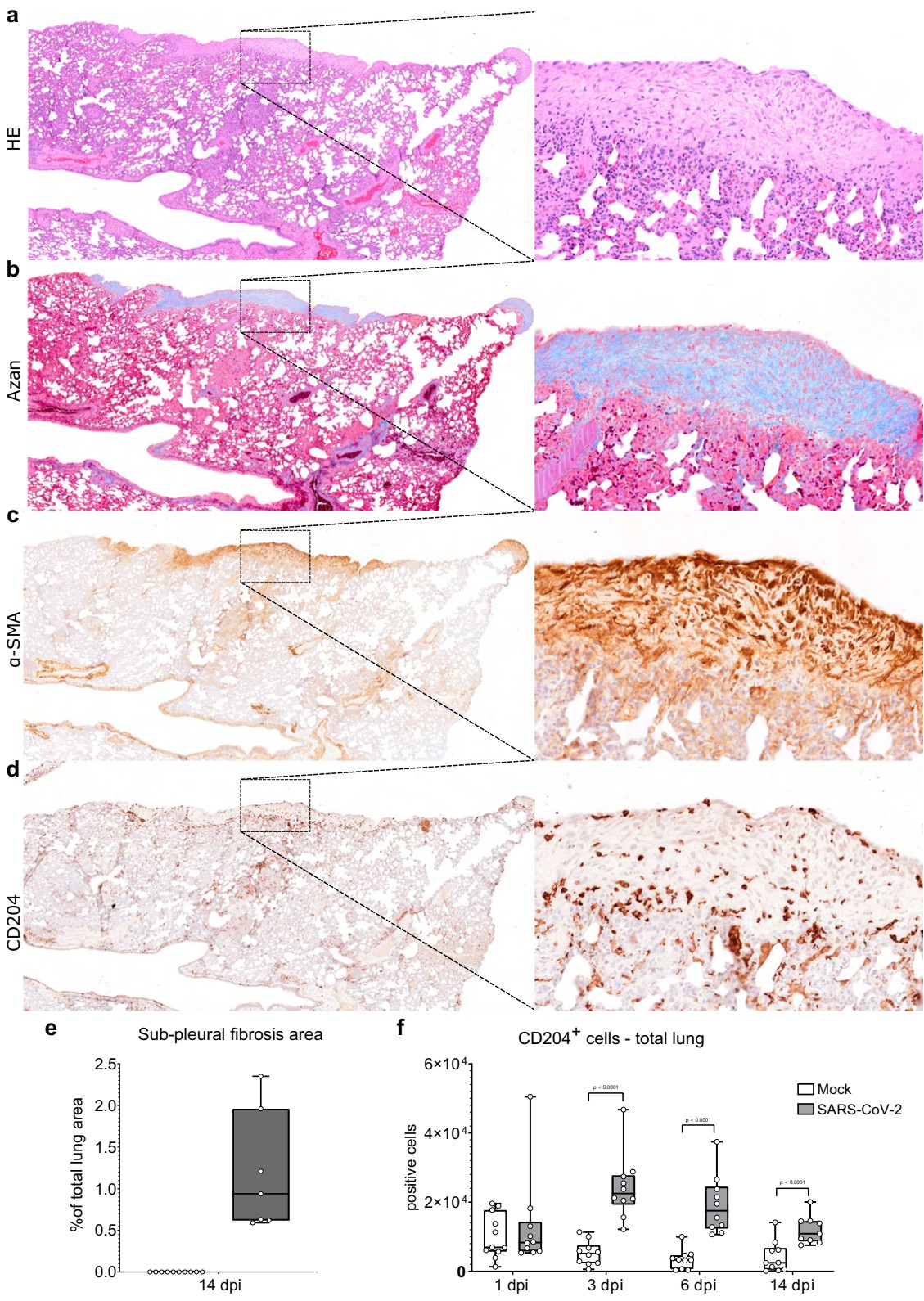

distribution. High module scores for ADI cell markers were detected throughout clusters 2 and 6 and partly in cluster 5, which were the clusters that showed a double expression of *Rtkn2* and *Lamp3* (Fig. 7c). Interestingly, clusters showing high module scores for AT2 cell markers also partly showed high module scores for club cell markers (cluster 1, 3 and 5, Fig. 7d) and ciliated cell markers (Cluster 1, Fig. 7e). Interestingly, clusters 1 and 4 which were previously noted as unknown, showed high module scores for AT2,

club and ciliated cell markers or only AT2 cell markers, respectively.

At 14 dpi, high module scores for AT1 cell markers were seen in clusters 2, 5, and partly in cluster 7, in line with *Rtkn2* expression (Fig. 7f). High module scores for AT2 cell markers were found in clusters 1, 4, 6 and 8 (Fig. 7g), partially in line with *Lamp3* distribution. High module scores for ADI cell markers were detected in clusters 2, 5, 7, 9, and partly in clusters 0 and 3 (Fig. 7h). Clusters showing high

**Fig. 5 | Subpleural fibrosis in SARS-CoV-2 infected hamsters. a–d** Representative images showing sub-pleural fibrotic foci in a lung lobe of a SARS-CoV-2 infected hamster at 14 days post infection (dpi). The left panel shows an overview of one right lung lobe displaying multifocal, extensive, well-demarcated areas of sub-pleural fibrosis. The right panel shows at higher magnification of the area delineated by the rectangle. On hematoxylin-eosin (HE) stained sections, this lesion is characterized by sub-pleural aggregates of spindle cells and abundant, pale eosinophilic, fibrillary, extracellular matrix (**a**). Azan stain demonstrates the presence of mature collagen fibers in the matrix (blue signal, **b**). Immunohistochemistry shows abundant α-smooth muscle actin (α-SMA)⁺ myofibroblasts (brown signal in **c**) as well as infiltration with CD204⁺ M2 macrophages (brown signal, **d**). **e** Quantification of sub-pleural fibrosis in lungs of mock and SARS-CoV-2 infected hamsters at 14 dpi. The percentage of affected area relative to total lung area is given. **f** Quantification of CD204⁺ M2 macrophages in total lung area. Data are shown as box and whisker plots. The bounds of the box plot indicate the 25th and 75th percentiles, the bar indicates medians, and the whiskers indicate minima and maxima. Dots indicate individual values. Data from CD204 quantification was tested by two-tailed Mann−Whitney $U$ test. A $p$ value of ≤0.05 was chosen as the cutoff for statistical significance. $N = 10$ animals/group for mock and SARS-CoV-2 respectively. Source data are provided as a Source Data file. Scale bars (**a**−**d**): 500 µm (overview in **a**−**d**), 100 µm (high magnification in **a**−**d**).

module scores for AT2 cell markers also partly showed high module scores for club cell markers (cluster 1, 4, 6, and 8 Fig. 7i). Interestingly, cluster 1 that was previously noted as unknown, showed high module scores for AT2 and club cell markers. Other unknown clusters like cluster 0, 3, and 9, showed high module scores for ADI cell markers. In addition, clusters 2, 5, 7 previously showed a double expression of *Rtkn2* and *Lamp3*, showed also high module scores for ADI cell markers. Cluster 10, previously unknown, separated completely from the other populations and showed a high score for ciliated cell markers (Fig. 7j).

Since our aim was to identify ADI cells, we also investigated the mean expression levels of all individual genes in the ADI cell marker list in each cluster (Supplementary Table 1) to support the module score analysis results. The bar plots with statistical analysis and details on gene expression within clusters are given in Supplementary Fig. 28, 29 and Supplementary Data 3–5. At 5 dpi, clusters 2 and 6 displayed significantly high mean expression levels for at least 8 ADI genes (cluster 2: 8/28 genes; cluster 6: 19/28 genes; Supplementary Fig. 28 and Supplementary Data 3 and 4). Cluster 5 did not show significantly high mean expression levels for any of the investigated ADI genes. These results partly confirmed the module score results and suggest that both AT1 and ADI cells belong to clusters 2 and 6.

At 14 dpi, clusters 0, 2, 3, 5, 7, and 9 displayed significantly high mean expression levels for at least 3 ADI genes (cluster 0: 6/28 genes; cluster 2: 10/28 genes; cluster 3: 3/28 genes; cluster 5: 9/28 genes; cluster 7: 19/28 genes; cluster 9: 9/28 genes; Supplementary Fig. 29 and Supplementary Data 3 and 5). These results confirmed the module score results and suggest that both AT1 and ADI cells belong to clusters 2, 5, and 6 as well as ADI cells belong to clusters 0, 3, and 9.

Taken together, transcriptome analysis identified clusters showing high module scores for AT1, AT2, ADI, club, and ciliated cell marker gene sets. Additionally, we identified clusters with significantly higher expression levels of ADI marker genes compared to other clusters. We interpret these differences in expression levels as the absence or presence of the respective cell populations. Based on this, we assigned one or more presumable cell identities for each cluster (Fig. 8a, b). We postulate that ADI cells showed an overlap of gene expression with AT1 cells, particularly at 5 dpi. At 14 dpi, a more distinct ADI cell population could be detected, which separated from the AT1 and AT2 clusters.

Next, we wanted to investigate if cell clusters with a high expression of ADI genes showed high module scores for pathways involved in lung regeneration. For this, we performed module score analysis with selected hallmark gene lists (http://www.gsea-msigdb.org/gsea/msigdb/index.jsp): p53 pathway, DNA repair, TGF-β signaling, notch signaling, wnt / β catenin signaling, epithelial−mesenchymal transition (EMT), angiogenesis, and G2M checkpoint. As described above, ADI cells in mice and humans express Tp53 and other markers of cell cycle arrest and DNA repair. The transcriptome data showed that two clusters with high ADI/AT1 module scores showed high positive scores for p53 pathway genes at 14 dpi (Fig. 8c, d). At 5 dpi, almost all clusters showed positive scores for DNA repair genes, with the highest scores observed in cells within clusters with high AT1/ADI

module scores (Fig. 8e). At 14 dpi, mainly clusters with high AT1/ADI and ADI module scores displayed positive scores for DNA repair (Fig. 8f). The AT2-ADI-AT1 trajectory is regulated by different signaling pathways, including TGF-β-, notch - and wnt/ β-catenin signaling and involves the EMT process[19,52]. At 5 dpi, a cluster with high AT2/club cell module scores and a cluster with high AT1/ADI cell module scores, partly showed high positive scores for TGF-β signaling. At 14 dpi, high positive scores for this gene set were partly detected in one cluster with high scores for AT1/ADI (Fig. 8g, h). A cluster with high AT1/ADI module scores partly revealed high positive scores for wnt/β-catenin signaling hallmark genes at 5 dpi (Fig. 8i). At 14 dpi, mainly clusters with high AT1/ADI and ADI cell module scores showed positive scores for wnt/ β-catenin signaling hallmark genes (Fig. 8j). In addition, clusters with high AT2/club cell module scores also partly showed high scores for wnt/β-catenin signaling hallmark genes at 14 dpi (Fig. 8j). A cluster with high AT2/club cells module scores revealed a high positive score for notch signaling hallmark genes at 5 dpi, whereas variably positive scores were distributed among clusters with high AT2/club cell, ADI and AT1/ADI cell module scores at 14 dpi (Fig. 8k, l). High positive scores for EMT hallmark genes were mainly found in clusters with high AT1/ADI module scores and partly within a cluster with high AT2/club cell module scores at 5 dpi (Fig. 8m). At 14 dpi, a cluster with high AT1/ADI module scores showed positive scores for EMT hallmark genes (Fig. 8n). Subsequently, we investigated the expression of genes involved in angiogenesis, since this process is upregulated in late phases of DAD, in the context of fibrosis[53]. Clusters with high AT1/ADI module scores at 5 dpi and clusters with high AT1/ADI or ADI module scores at 14 dpi revealed high positive scores for angiogenesis hallmark genes (Fig. 8o, p). Finally, we wanted to investigate which cell clusters show high scores for G2M checkpoint hallmark genes as a readout of cell proliferation. The highest scores for this gene set were observed in clusters with high AT2/club cell and AT1/ADI cell module scores at 14 dpi (Fig. 8q, r).

## Discussion

The COVID-19 pandemic has claimed many lives and challenged the global healthcare system in an unprecedented way. Survivors of acute disease may be faced with a wide spectrum of long-lasting symptoms, with pulmonary, neuropsychiatric, and cardiovascular sequelae at the forefront, which have a negative impact on the quality of life. Considering the staggering amount of patients reporting prolonged symptoms even as long as 15 months after the initial onset of COVID-19[2,7,54,55], further research into potential pathomechanisms of this protracted recovery is urgently needed[56]. A possible explanation for the mechanisms underlying some PASC symptoms, such as dyspnea, shortness of breath, and exercise intolerance, could be an impaired regeneration of alveolar tissue and lung fibrosis[3,9]. It has also been suggested that the persistence of CK8⁺ ADI cells might be the cause of prolonged hypoxemia in COVID-19 patients[22]. Importantly, these conclusions are based on observations from tissues collected from acute, lethal COVID-19 cases. In contrast, we can only speculate about the presence of these cells in PASC, since samples from affected humans are scarce. Therefore, the establishment and

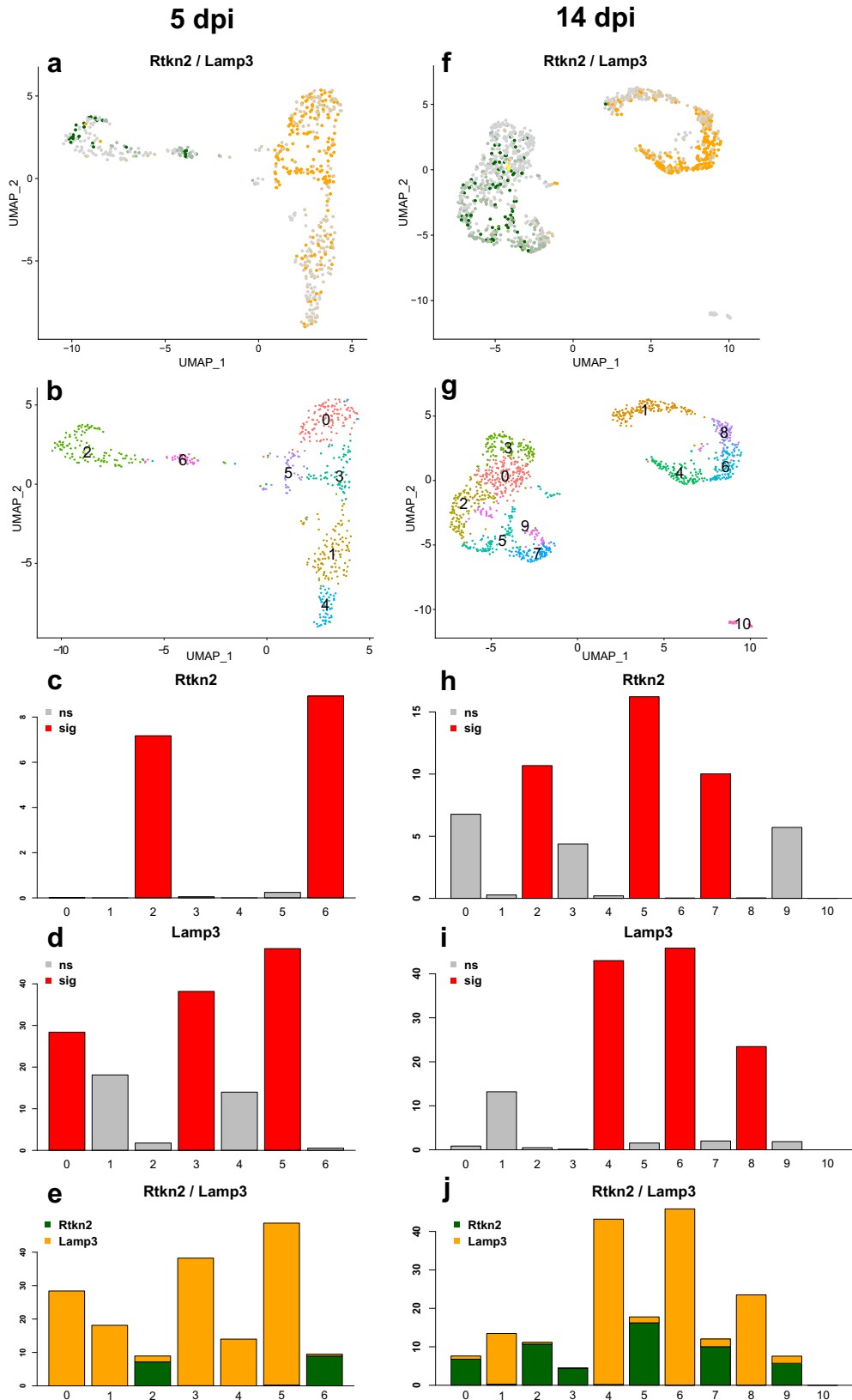

further characterization of appropriate animal models of PASC are urgently needed.

SARS-CoV-2 infected hamsters reliably phenocopy moderate to severe COVID-19[29]. Recovering hamsters show a pronounced epithelial cell proliferation within airways and alveoli, which started at 3 dpi and was detectable until 14 dpi in the present study. This is in line with

previous reports, which showed that proliferative foci can persist up to 31 dpi in hamsters[35,57]. Here, we characterized the proliferating epithelial cell types in more detail. First, we investigated the AT2-ADI-AT1 trajectory on a morphologic level. In mouse models of lung injury, the transition from AT2 to the ADI state is characterized by a progressive decrease of cell sphericity, expression of CK8, and loss of AT2 marker

**Fig. 6 | Single-cell RNASeq analysis of alveolar cells in SARS-CoV-2 infected hamsters.** Single-cell RNA-Seq data set (GSE162208) from lungs of SARS-CoV-2 infected hamsters at 5 (**a**–**e**) or 14 (**f**–**j**) days post infection (dpi). **a**, **f** Expression of alveolar pneumocyte type 1 (AT1) marker gene *Rtkn2* (green) and alveolar pneumocyte type 2 (AT2) marker gene *Lamp3* (orange). **b**, **g** Cells were re-clustered at 5 and 14 dpi resulting in 7 and 11 clusters, respectively. **c** *Rtkn2* mean expression level among clusters at 5 dpi. **d** *Lamp3* mean expression level among clusters at 5 dpi. **e** *Rtkn2* and *Lamp3* mean expression level among clusters at 5 dpi. **h** *Rtkn2* mean expression level among clusters at 14 dpi. **i** *Lamp3* mean expression level among clusters at 14 dpi. **j** *Rtkn2* and *Lamp3* mean expression level among clusters at 14 dpi. Bar plots indicate mean expression, **c**, **d**, **h**, **i** red: significant mean gene expression (multiple testing adjusted *p* value < 0.05); gray: not significant mean gene expression, *y* axis: mean gene expression level; *x* axis: cluster number.

expression[19–22]. Double-labeling of SPC and CK8 demonstrated the transition of AT2 to ADI cells, associated with phenotypical changes as described above in SARS-CoV-2-infected hamsters. At 6 dpi, all stages of ADI cells were observed, including round, SPC⁺CK8⁺ cells (early ADI stage) and polygonal, plump to elongated, SPC⁻CK8+ cells (late ADI stage). Interestingly, at 14 dpi, we observed numerous round, SPC⁺CK8⁺, early ADI stages, and fewer late ADI stages, which could indicate a new wave of ADI cell generation at this time-point. Lineage-tracing studies in the mouse bleomycin lung injury model demonstrated that ADI cells could develop from AT2 as well as from MHCII⁺ club cells migrating from the airways[19]. In the early stages after injury, peaking at 5 dpi, ADI cells are mainly derived from AT2 cells, while club cell-derived ADI cells appear later, peaking at 10 dpi. Of note, a part of the MHCII⁺ club cells differentiating towards ADI cells goes through an SPC⁺ stage[19]. We speculate, that the round SPC⁺CK8⁺ ADI cells observed at 14 dpi in SARS-CoV-2 infected hamsters could be derived from airway progenitors analogous to murine MHCII⁺ club cells, which transiently assume an AT2 stage.

In addition to the demonstration of transitional cell stages on a morphological level, the presence of ADI cells in the hamster model of COVID-19 was supported using scRNA-seq transcriptome data from a previously published study. The results showed a partial overlap in gene expression between ADI and AT1 cells with a variable separation of these cell types depending on the time point after infection. CK8⁺ADI cells in SARS-CoV-2 infected hamsters frequently expressed nuclear TP53 protein. Transcriptome data also showed that some clusters with high ADI gene expression displayed high module scores for p53 pathway and for DNA repair hallmark genes. Nuclear TP53 regulates the transcription of genes involved in cell cycle arrest and DNA repair and accumulation of TP53 is therefore detected in cells with high level of DNA damage[58]. ADI cells undergo mechanical stretch-induced DNA damage while migrating to cover the denuded septa and to differentiate into AT1[21,59]. The nuclear expression of TP53 could reflect a particularly high level of injury, triggering DNA repair mechanisms. It is important to underline that in SARS-CoV-2 infected hamsters, nuclear TP53 expression was often found in hypertrophic CK8⁺ cells with a bizarre morphology, binucleation, or karyomegaly. We assume that these hypertrophic cells have accumulated a high level of DNA damage, are blocked in the ADI stage, and are not likely to differentiate into slender AT1 cells. A permanent block in the ADI cell state has been described in idiopathic pulmonary fibrosis (IPF) and mouse models of lung fibrosis[20,22,60,61]. Importantly, it has been demonstrated in a mouse model that induction of TP53-dependent AT2 senescence is sufficient to propagate progressive pulmonary fibrosis[52,61]. Besides TP53, other signaling pathways have been implicated in ADI cell senescence. For instance, in vitro studies in primary murine cells revealed that chronic activation of wnt/β-catenin signaling can induce senescence and CK8 expression in ADI cells[52,62]. In addition, persistent Notch activation in AT2 cells induces retarded differentiation of AT2 into AT1 cells, resulting in ADI cell accumulation in a *Pseudomonas* lung injury model[52,63]. Moreover, persistent TGF-β signaling has been shown to block ADI cells from differentiating into AT1 cells[20]. We showed that, from 5 to 14 dpi, an increasing number of clusters with high ADI gene expression showed high scores for wnt/β-catenin and notch signaling hallmark genes, while module scores for TGF-β signaling pathway were less prominent and detected only in one cluster. Therefore, we speculate that prolonged wnt/β-catenin and/or

notch signaling, rather than excessive TGF-β, could be responsible for the prolonged presence of ADI cells in SARS-CoV-2 infected hamsters. However, the available data do not allow us to assess the duration of the activation of the respective pathways in ADI cells, and further studies with a more detailed analysis and additional time points are warranted to confirm this hypothesis. Besides the dysregulation of the discussed pathways, a direct contribution of viral infection to the induction of senescence must be considered. It has been demonstrated that SARS-CoV-2 and other viruses can induce cellular senescence in infected AT2 cells[64].

The clinical relevance of the observed ADI cell accumulation in hamsters deserves further investigation. In COVID-19 patients with a severe disease course and lethal outcome, high numbers of ADI cells were detected by others and in the present study, which indicates that dysregulated alveolar regeneration could play a role in the pathogenesis of severe disease[9,22]. In line with this, we found that TP53 is expressed by CK8⁺ ADI cells in lethal COVID-19 samples, but not in CK8⁺ ADI cells in a non-COVID pneumonia case.

In addition to the presence of ADI cells, the majority of SARS-CoV-2 infected animals showed foci of sub-pleural fibrosis at 14 dpi, indicative of irreversible damage/remodeling. This is in line with previous reports in hamsters[65,66]. The pattern of fibrosis is similar to what has been described in IPF patients and a RhoGTPase Cdc42 deletion mouse model of progressive pulmonary fibrosis[24,67]. In these conditions, a progression of fibrotic lesions from periphery to center is typically encountered[19,24]. Subpleural alveoli are subject to increased mechanical tension during respiration, which has been shown to activate TGF-β-mediated pro-fibrotic processes[24,67]. In addition to fibrotic foci, our study also revealed a prominent presence of CD204⁺ M2 macrophages starting at 3 dpi and persisting until 14 dpi. M2 macrophages are known to promote fibrosis by a variety of factors, including TGF-β secretion[68]. Thus, the fibrosis could be promoted by the prolonged presence of an unfavorably polarized inflammatory response. In addition to macrophages, AT2 cells can promote a pro-fibrotic microenvironment by activating local fibroblasts to become myofibroblasts via paracrine signaling, as demonstrated in vitro[53,69,70]. This process was initiated by an induction of an EMT process in the AT2 cells[61,70]. Of note, it has been reported that ADI cells can display high scores for EMT genes[19] and the results from our transcriptome analysis showed that cells in clusters with AT1/ADI gene expression also showed high EMT pathway module scores at 5 and 14 dpi in SARS-CoV-2 infected hamsters. However, an EMT process with conversion of ADI cells into fibroblasts or other mesenchymal cells could not been demonstrated in murine models of lung injury[18–21]. Thus, it has been postulated that upregulation of genes typically involved in this process is necessary for the morphological changes that ADI cells undergo completing the AT2-ADI-AT1 trajectory (i.e., flattening and spread), rather than entering a mesenchymal-like cell state[19]. In addition, it has been reported that lung fibrotic lesions in COVID-19 patients are preceded by a prolonged blood vessel neoformation[71]. Interestingly, transcriptome data revealed that clusters with high AT1/ADI and ADI gene expression showed high positive scores for angiogenesis hallmark genes at 14 dpi, suggesting that ADI cells in hamsters might contribute also to a pro-angiogenetic microenvironment, promoting vascular changes during lung fibrosis similar to COVID-19 patients. A recent study in a mouse model of COVID-19 demonstrated that aged mice infected with a mouse-adapted strain of SARS-CoV-2 show fibrotic lesions starting from 15 dpi and persisting up

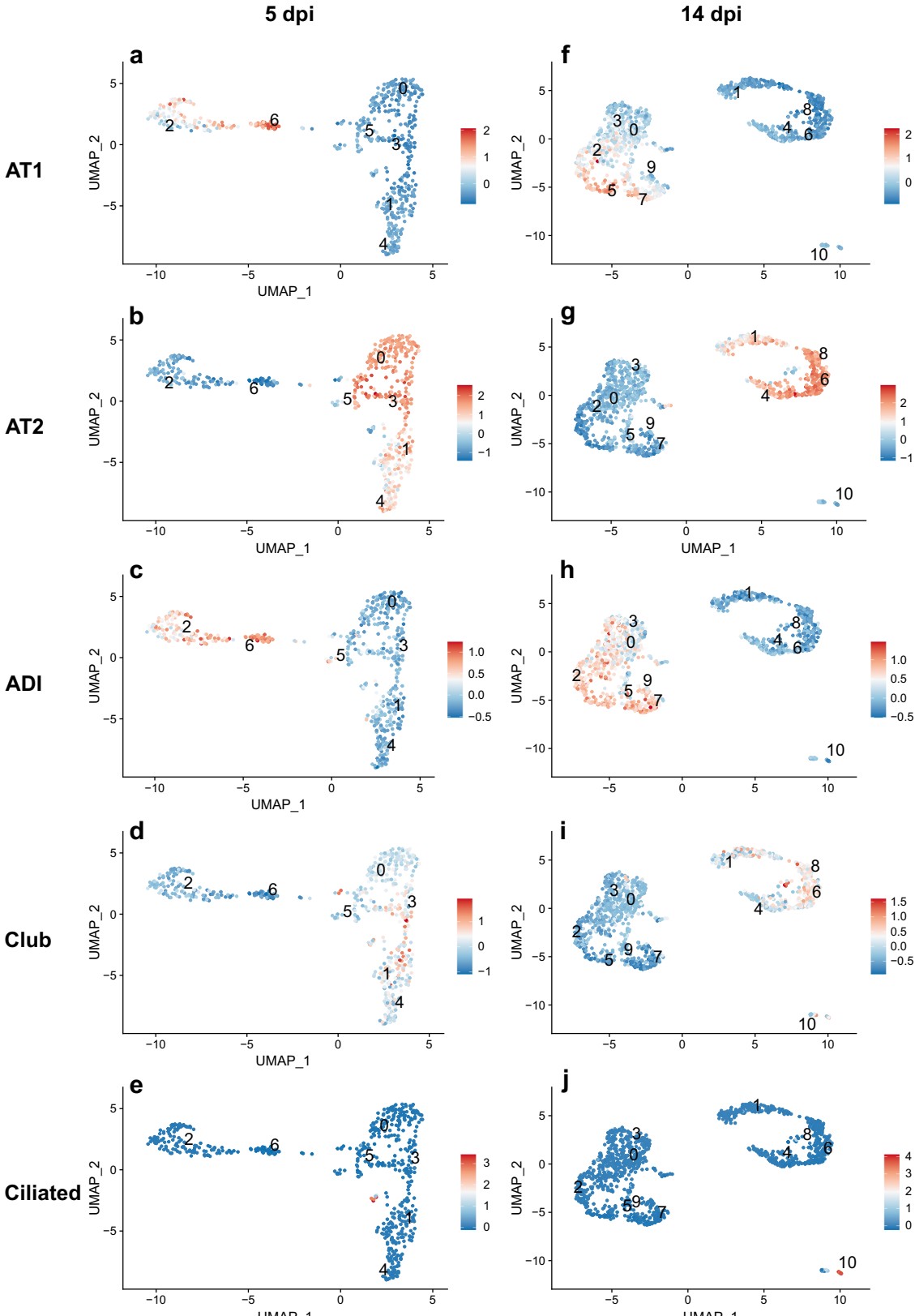

**5 dpi**

**14 dpi**

**Fig. 7 | Module scores for cell marker gene sets within alveolar cells in SARS-CoV-2 infected hamsters. a–j** Results from module score analysis for alveolar pneumocytes type 1 (AT1), alveolar pneumocytes type 2 (AT2), alveolar differentiation intermediate (ADI), club, and ciliated cell marker gene sets at 5 (**a**–**e**) and 14 (**f**–**j**) days post infection (dpi). Cell types are indicated in the figure, labels on the left. For cell marker gene list, see Supplementary Table 1. For individual gene feature and ridge plots see Supplementary Fig. 8–27. For individual ADI cell genes bar plots showing mean expression levels and significances for individual clusters, see Supplementary Data 3–5 as well as Supplementary Figs. 28, 29.

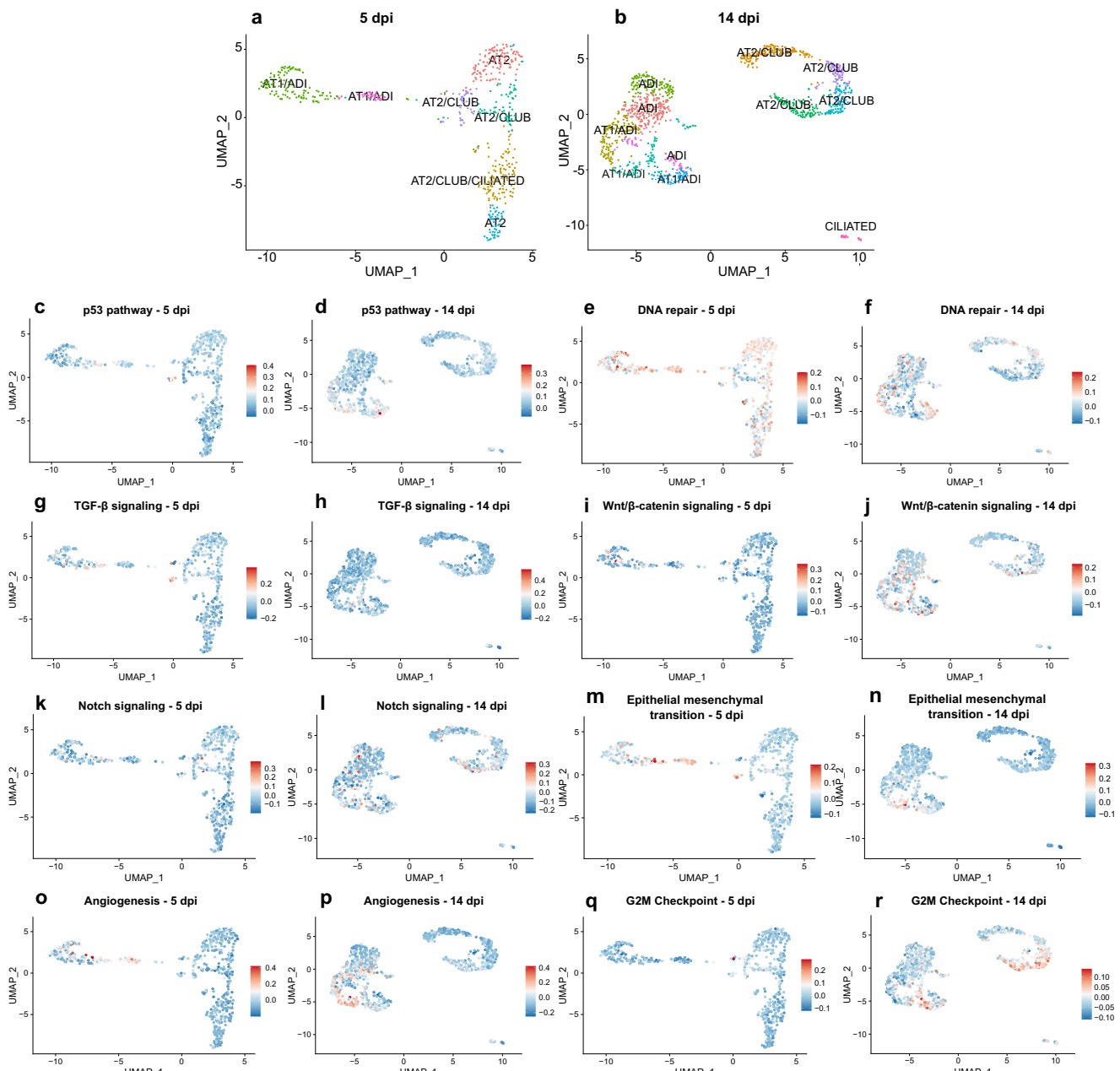

**Fig. 8 | Module scores for GSEA hallmark genes within alveolar cells in SARS-CoV-2 infected hamsters. a, b** cluster names were assigned based on cell marker genes module scores results. **c–r** Results from module score analysis for GSEA hallmark genes.

to 120 dpi[60]. Similar to what we observed in the hamster model, the lesions were characterized by a subpleural deposition of collagen and the presence of α-SMA-positive myofibroblasts. The authors also described elevated numbers of M2-type macrophages, which persisted in chronic lesions. Moreover, this study also analyzed the dynamics of AT2-derived ADI cells and demonstrated that persistence of ADI cells is a feature of chronic lesions, in line with our findings in the hamster model. However, the study from Dinnon et al.[60] did not investigate airway progenitor cell contribution to alveolar regeneration. In contrast to this, we found a prominent airway progenitor mobilization into damaged alveoli in our hamster model, indicating that hamsters model this aspect of lung regeneration observed in humans more closely than mice.

Although pre-existing AT2 cells are described to be the predominant source of AT1s after alveolar damage, it is known that other cell types partake in regenerative processes, especially after severe injury[15,42]. In case of severe damage that involve broad epithelial denudation, basal cells can migrate into alveoli, become distal basal-like cells and subsequently promote alveolar regeneration giving rise to AT2[15,40,52]. A contribution of airway progenitors to alveolar repair has been reported in COVID-19 patients[25,72]. In COVID-19 patients, the most prominent airway progenitors supporting alveolar regeneration were reported to be CK5[+] basal cells, which form the so-called "keratin 5 pods". Basal cell expansion, also termed "pod" is gradually recognized as a common feature of epithelial remodeling[17,52]. To a lesser extent, more immature CK5[+]p63[+] basal cells were also reported to support alveolar regeneration in COVID-19 patients[40]. Conversely, in SARS-CoV-2 infected hamsters, we found predominantly CK14[+] cells within alveolar proliferation foci, resembling the human CK5[+] pods. Basal airways cells originate from the same CK5[+]CK14[+]p63[+] pool that gives rise to different combinations of CK5[+/–], CK14[+/–], p63[+/–] progenitor cells that will populate the airways[43]. Some of these cells also have the

potential to give rise to AT2 cells[42,43]. It appears that the subpopulation might differ among various species. Human lung multipotent cells can differ from murine ones, and in its turn, we might expect the same for other rodents like hamsters. The CK14$^+$ basal cells detected in our study were having similar features like the ones described for human CK5$^+$ cells, namely pods formation and differentiation towards AT2 cells, and therefore can be considered the hamster equivalent of human basal cells contributing to alveolar repair.

The authors recognize that the study has some limitations. First, this work provides a whole slide digital quantification of the main cell types involved in alveolar regeneration upon SARS-CoV-2 infection, including CK8$^+$ ADI cells. However, since that several tested antibodies (anti-AGER, -AQP5, -PDPN) failed to specifically recognize AT1 cells in hamsters, quantification of these cells and demonstration of ADI-AT1 transition by double-labeling was not possible. Therefore, the ADI-AT1 transition was demonstrated with ultrastructural analysis, in line with previous COVID-19 reports. Second, we can only speculate on the clinical relevance of ADI persistence and fibrotic lesions in animals. However, once this work confirmed hamsters to be a reliable model for these features, further investigations including longer time points and the assessment of lung function and gas-exchange capacity are warranted. Third, the conclusions regarding cell origins of ADI as well as cell fate of CK14$^+$ basal cells and AT2 cells in this work are based on double-labeling and co-expression of genes interpreted in the context of published data and therefore have to be interpreted as preliminary observations. Additional studies involving lineage-tracing are required to confirm our interpretation. Fourth, the conclusions from transcriptome analysis require confirmation in future studies. It is possible that expression of certain genes may not have been detected because of limited cell numbers or depth of sequencing. It is thus possible that cells with lower expression levels or module scores are also present in other clusters. In addition, future studies including evaluation of protein expression of genes involved in the analyzed pathways are needed also to confirm our interpretations. In conclusion, our study provides a detailed characterization of cell populations composing the pulmonary epithelial regenerative response in the hamster and thus provides preliminary and highly needed information about this important translational COVID-19 model. We show that ADI cells and airway-derived progenitors participate in alveolar regeneration in the species and provide evidence of ongoing regeneration post virus clearance. Thus, hamsters are a suitable model to investigate the relevance of these changes and their actual contribution to PASC symptoms. However, further studies including a longer investigation period and more detailed clinical analyses are required. Since post-COVID-19 pathological lesions show overlap with other diseases featuring DAD and IPF, the model can be used for broader implications.

## Methods

### Ethics statement
The animal experiment was in accordance with the EU directive 2010/63/EU and approved by "Behörde für Justiz und Verbraucherschutz der Freien und Hansestadt Hamburg, Abteilung für Lebensmittelsicherheit und Veterinärwesen" (protocol code N032/2020 22 April 2020). For the human samples, all patients or their relatives provided written informed consent for the use of their data and samples obtained during autopsy for scientific purposes. Ethical approval was given by the local institutional review board at Hannover Medical School (no. 9621_BO_K_2021).

### Hamster study
During the experiment, the animals were under veterinary observation and all efforts were made to minimize distress. Eight to ten weeks old male and female Syrian golden hamsters (*Mesocricetus auratus*) purchased from Janvier Labs were housed under BSL-3 conditions for 2 weeks prior to the experiment for acclimatization. A total of 80

hamsters divided into groups of 5 male and 5 female ($n = 10$) animals per time point per infection group were housed in isolated ventilated cages under standardized conditions ($21 \pm 2\,°C$, 40–50% relative humidity, 12:12 light-dark cycle, food, and water ad libitum) at the Leibniz Institute for Experimental Virology in Hamburg, Germany. Animals were infected with an intranasal inoculation of either a suspension containing $10^5$ plaque-forming units (pfu) of SARS-CoV2 (SARS-CoV-2/Germany/Hamburg/01/2020; ENA study PRJEB41216 and sample ERS5312751) or phosphate-buffered saline (PBS, control) as previously described[73] under general anesthesia. At 1, 3, 6, and 14 days post-infection (dpi), groups of five female and five male hamsters ($n = 10$) per each treatment (either SARS-CoV-2 infected or mock-infected) were euthanized by intraperitoneal administration of a pentobarbital-overdose and blood withdrawal by cardiac puncture. Immediately after death, right lung lobes (*lobus cranialis, lobus medius, lobus caudalis, lobus accessorius*) were collected and fixed in 10% neutral-buffered formalin (Chemie Vetrieb GmbH & Co) or 5% glutaraldehyde (Merck KGaA) for microscopic and ultrastructural evaluation respectively. For this work, no sex-based analysis has been performed, since only five males and 5 females were present in each group. These low numbers would not have allowed a reliable statistical analysis. For this reason, both sexes were pooled.

### Virus
SARS-CoV-2/Germany/Hamburg/01/2020 is a SARS-CoV-2 ancestral strain and was isolated from a nasopharyngeal swab of a confirmed COVID-19 patient. Stock virus was produced after three serial passages in Vero E6 cells using Dulbecco's Modified Eagle's Medium (DMEM; Sigma) supplemented with 2% fetal bovine serum, 1% penicillin-streptomycin and 1% L-glutamine at 37 °C. The infection experiment was carried out under biosafety level 3 (BSL-3) conditions at the Leibniz Institute for Experimental Virology in Hamburg, Germany. Information about the virus genome sequence are available here: European Nucleotide Archive (ENA), study: PRJEB41216, sample: ERS5312751, and accession number: SAMEA7556109).

### Human samples
Lung samples were obtained from three patients who died of respiratory failure caused by severe COVID-19. The patients were two men, aged 76 and 74 years, and one woman, aged 74 years. The patients were hospitalized for 21, 7, and 5 days, respectively, and all received mechanical ventilation. SARS-CoV-2 infection was confirmed by PCR. The lung samples were obtained during autopsy. In addition, one non-COVID-19 lung sample was obtained from a 66-year-old man who underwent a lobectomy due to a pulmonary neoplasm. Man/women refers to sex, gender information was not available. Sex was not considered in the selection process. No quantitative analysis was performed with human samples, and therefore, no source data or sex-based analysis is provided.

### Histopathology
For histopathological evaluation, lung samples were formalin-fixed and embedded in paraffin. Serial sections of 2µm were cut and stained with hematoxylin and eosin (HE) and Azan trichrome. Qualitative evaluations with special emphasis on inflammatory and epithelial regenerative processes (HE) as well as on fibrosis (Azan) were performed in a blinded fashion by veterinary pathologists (FA, LH) and subsequently reviewed by board-certified veterinary pathologist (MCI, WB).

### Immunohistochemistry
Immunohistochemistry was performed to detect SARS-CoV-2 antigen (SARS-CoV-2 nucleoprotein), macrophages and dendritic cells (ionized calcium-binding adapter molecule 1, IBA-1), alveolar pneumocytes type 2 (pro-surfactant protein C), alveolar differentiation intermediate cells

(cytokeratin 8), airway basal cells (cytokeratin 14), club cells (secretoglobin 1A1), and M2 macrophages (CD 204). Immunolabelings were visualized either using the Dako EnVision+ polymer system (Dako Agilent Pathology Solutions) and 3,3′-Diaminobenzidine tetrahydrochloride (DAB, Carl Roth) as previously described[36] or using avidin–biotin complex (ABC) peroxidase kit (Vector Labs) and DAB (Carl Roth) as previously described[74]. Nuclei were counterstained with hematoxylin. Further details about primary and secondary antibodies, visualization methods and dilutions used can be found in Supplementary Table 2. For negative controls, the primary antibodies were replaced with rabbit serum or BALB/cJ mouse ascitic fluid, respectively, with the dilution chosen according to protein concentration of the exchanged primary antibody. Antibodies were tested on murine and human lung tissue to confirm specificity for the cells of interest. Subsequently, murine and human tissues were used as positive controls.

### Immunofluorescence

Double-labeling immunofluorescence was performed to investigate different states of alveolar pneumocytes type 2 and alveolar differentiation intermediate cells, as well as to prove that airway progenitor cells can differentiate into alveolar cell types. The reaction was carried out as previously described with minor modifications[75]. Briefly, after deparaffinization, HIER, and serum blocking, washing with PBS in between each step, a dilution containing two primary antibodies was added and incubated overnight at 4 °C. Afterwards, a dilution containing two secondary antibodies was incubated for 60 minutes at room temperature in the dark. After washing with PBS and distilled water, sections were counterstained and mounted using anti-fade mounting medium containing DAPI (Vectashield®HardSet™, Biozol). Further details about primary and secondary antibodies, visualization methods and dilutions used can be found in Supplementary Table 3. For negative controls, the primary antibodies were replaced with rabbit serum or BALB/cJ mouse ascitic fluid respectively with the dilution chosen according to protein concentration of the exchanged primary antibody.

### Transmission electron microscopy (TEM)

In order to detect AT1 cells with features of AT2 proving the final trajectory ADI-AT1 in hamsters, transmission electron microscopy was performed. Reactions were carried out as previously described[73,76]. Briefly, glutaraldehyde-fixed lung tissue was rinsed overnight in cacodylate buffer (Serva Electrophoresis GmbH), followed by post-fixation treatment in 1% osmium tetroxode (Roth C. GmbH & Co. KG). After dehydration using a graded alcohol series, samples were embedded in epoxy resin. Representative areas of affected alveoli were then cut into ultrathin sections, contrasted with uranyl acetate and lead acetate, and subsequently morphologically evaluated using a transmission electron microscope (EM 10 C, Carl Zeiss Microscopy GmbH).

### Digital image analysis

To quantify immunolabeled cells in pulmonary tissue, areas of alveolar epithelial proliferation as well as areas of subpleural fibrosis, slides were digitized using an Olympus VS200 Digital slide scanner (Olympus Deutschland GmbH). Image analysis was performed using QuPath (version 0.3.1), an open-source software package for digital pathology image analysis[77]. For all animals, whole slide images of the entire right lung were evaluated. For the pro-surfactant protein C (proSP-C), cytokeratin 8 (CK8), cytokeratin 14 (CK14), secretoglobin 1A1 (SCGB1A1) immunolabelings, total lung tissue was first detected automatically using digital thresholding. Afterwards, regions of interest (ROI) were defined. The ROIs "airways" (bronchi, bronchioli, terminal bronchioli), "blood vessels", "affected alveoli" (alveoli that were involved either in an inflammatory process or in an epithelial regenerative process or both), and "artifacts" were manually outlined.

The area denoted as "total alveoli" was defined by subtraction of the "blood vessels", "airways" and "artifacts" ROIs from the total lung tissue using an automated script. The area denoted as "unaffected alveoli" (alveoli that were morphologically free from any inflammatory or regenerative process) was defined by subtracting the ROI "affected alveoli" from the ROI "total alveoli" using an automated script. Using tissue- and marker-specific thresholding parameters, quantification of immunolabeled cells was achieved by automated positive cell detection in all ROI. To analyze SARS-CoV-2 NP, IBA-1, and CD204 immunolabeling, total lung tissue was automatically detected using digital thresholding. Afterwards, only blood vessels and artifacts were indicated as ROIs and subtracted from the total lung tissue. Based on tissue and marker-specific thresholding parameters, quantification of immunolabeled cells was then achieved by automated positive cell detection. For quantification of alveolar epithelial proliferation or subpleural fibrosis, total lung tissue area was automatically detected using digital thresholding. Subsequently, either alveolar epithelial proliferation or subpleural fibrosis were marked as ROIs and the total area was calculated. Finally, the percentage of total lung area affected by either epithelial proliferation or subpleural fibrosis was obtained. All procedures (tissue detection, indication of ROIs, positive cell detection) were performed and subsequently reviewed by at least two veterinary pathologists (FA, GB, LH, MC). Statistical analysis and graphs design were performed using GraphPad Prism 9.3.1 (GraphPad Software, San Diego, CA, USA) for Windows™. Single comparisons between SARS-CoV-2 infected hamsters and the control group were tested with a two-tailed Mann–Whitney $U$ test. For multiple comparisons among different time-points data were tested for significant differences using Kruskal–Wallis tests and corrected for multiple group comparisons using the Benjamini–Hochberg correction. Statistical significance was accepted at exact $p$ values of ≤0.05.

### Single-cell RNAseq

Single-cell RNASeq data from lungs of SARS-CoV-2 infected hamsters were obtained from a publicly available dataset[45]. The dataset was selected based on a GEO search using the terms "hamster", "SARS", "lung", and "single-cell". The results were filtered by "organism" (*Mesocricetus auratus*) and "study type" (expression profiling by high throughput sequencing). The search yielded 11 studies, 8 of which included data on lung tissues from SARS-CoV-2 infected golden Syrian hamsters. From these, only one study (Nouailles et al. GSE162208) contained datasets from two different time-points, including a time-point after resolution of acute infection. Data were analyzed using the R software package (version 3.6.0)[78]. Expression data were downloaded from GEO (GSE162208) and Seurat objects (version Seurat_3.2.0)[46–49], were generated from downloaded h5 files by combining replicate samples from lungs at 5 and 14 dpi. Pre-processing of data was performed by applying several Seurat functions: subset (subset = nFeature_RNA > 200 & nFeature_RNA < 2500 & percent.mt <5), NormalizeData (data), FindVariableFeatures (data, selection.method = "vst", nfeatures = 2000), ScaleData(data, features = all.genes), and RunPCA(data, features = VariableFeatures (object = data)). Uniform Manifold Approximation and Projection (UMAP) were performed after clustering with Seurat functions FindNeighbors (pbmc, dims = 1:10) and FindClusters (pbmc, resolution = 0.5). AT1 and AT2 cell cluster were then identified by using the marker genes *Rtkn2* (AT1) and *Lamp3* (AT2), respectively, from the original publication[45]. These clusters were selected, then pre-processed and re-clustered as described above. We then collected more candidate marker genes for AT1, AT2, ADI cells and additional cell populations in these clusters by applying functions FindAllMarkers (pbmc, only.pos = TRUE, min.pct = 0.25, logfc.threshold = 0.25) and selecting the top 10 significant markers genes per cluster. We further identified additional candidate markers from[9,19,45]. We evaluated the specificity of these candidate markers by visualizing them with the functions FeaturePlot, RidgePlot,

and AddModuleScore of the Seurat package. The list of final maintained marker genes is presented in Supplementary Data 1. The function AddModuleScore of the Seurat package was used to visualize high module scores of the different cell populations and hallmark genes from the GSEA database [http://www.gsea-msigdb.org/gsea/msigdb][79]. Means in expression levels were calculated using function AverageExpression of the Seurat package, and differences in log-fold change and corresponding multiple testing adjusted *p* values were determined using function FindAllMarkers of the Seurat package[46–51].

## Statistic and reproducibility

The animal experiment, immunolabellings, quantifications, and data analysis were performed once. Sample size was determined with ANOVA and Kruskal–Wallis test. Allocation to experimental groups was randomized. No data were excluded from the analysis. The investigators were not blinded to group allocation during the animal experiment. For morphologic analysis, samples were blinded.

## Reporting summary

Further information on research design is available in the Nature Portfolio Reporting Summary linked to this article.

## Data availability

The data generated in this study are provided in the Supplementary Information and Source Data file. Single-cell RNASeq data from the lungs of SARS-CoV-2 infected hamsters were obtained from a publicly available dataset; reference from GEO is provided: GSE162208. Source data are provided with this paper.

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

## Acknowledgements

The authors are grateful to Julia Baskas, Petra Grünig, Jana-Svea Harre, Kerstin Rohn, Caroline Schütz, Kerstin Schöne, and Danuta Waschke for excellent technical assistance. This project was in part supported by the COVID-19 Research Network of the State of Lower Saxony (COFONI) with funding from the Ministry of Science and Culture of Lower Saxony, Germany (14-76403-184, project number 5FF22, F.A., M.C., W.B.). This research was in part supported by the Deutsche Forschungsgemeinschaft (DFG; German Research Foundation –398066876/GRK 2485/1, W.B., G.B., L.H.) This study was also supported in part by intra-mural grants from the Helmholtz-Association (Program Infection and Immunity), and NIAID Research Grants 2-U19-AI100625-06 REVISED and 5U19A|100625-07 (K.S.). This Open Access publication was funded by the Deutsche Forschungsgemeinschaft (DFG, German Research Foundation)—491094227 "Open Access Publication Funding" and the University of Veterinary Medicine Hannover Foundation.

## Author contributions

The study was designed by F.A., W.B., and M.C. The animal experiments were performed by S.S.-B., B.S., N.M.-K., S.B., M.Z., and G.G. Histology, immunolabelling, and electron microscopy evaluation of hamster tissues was conducted and analyzed by F.A., L.H., M.C., G.B., K.B., A.B., and W.B. Pathological analysis of human samples was performed by M.K. scRNA-seq analysis was performed by K.S. Data analysis and interpretation were performed by F.A., L.H., M.C., and G.B. Figures were prepared by M.C., K.S., and F.A. The original draft was written by L.H., M.C., F.A., and K.S. The manuscript was reviewed, edited, and approved by all authors. Funding was acquired by M.C., K.S., and W.B. The project was supervised by W.B. and F.A.

## Funding

## Competing interests

The authors declare no competing interests.

## Additional information

[1]Department of Pathology, University of Veterinary Medicine, Foundation, Hannover, Germany. [2]Department of Microbiology, Immunology and Biochemistry, University of Tennessee Health Science Center, Memphis, Tennessee, USA. [3]Institute of Virology Münster, University of Münster, Münster, Germany. [4]Department for Viral Zoonoses-One Health, Leibniz Institute for Virology, Hamburg, Germany. [5]Institute of Pathology, Hannover Medical School (MHH), Hannover, Germany. [6]Member of the German Center for Lung Research (DZL), Biomedical Research in Endstage and Obstructive Lung Disease Hannover (BREATH), Hannover Medical School (MHH), Hannover, Germany. [7]These authors contributed equally: Laura Heydemann, Małgorzata Ciurkiewicz, Wolfgang Baumgärtner, Federico Armando. ✉e-mail: Wolfgang.baumgaertner@tiho-hannover.de

