## [Peer review file Updated Version · Nature Communications]

REVIEWER COMMENTS

Reviewer #1 (Remarks to the Author):

The manuscript by Heydemann et al. provides a detailed histology-based description and characterization of alveolar regeneration following SARS-CoV-2 infection in hamsters. This appears to be the first such detailed description of this process in hamsters, has accompanying data from human autopsy specimens and has found publicly available transcriptomic data for a separate study that largely supports the observations of this study.

The only major criticisms would be that perhaps many of the findings are not unexpected. However, given that this is not previously thoroughly described, this data set appears to be novel. And PASC is characterized in humans at the >12 weeks mark after acute infection, but this study did not assess timepoints beyond 14 days post-infection, which in the hamster model is only 7 days beyond when virus can be detected. The authors do highlight this as a limitation and suggest that future work should address this.

Minor comments

It would appear that this work may have been part of another study. Reference to the study seems to be missing (if it is published). This would be helpful as there is no virus data included in this paper, apart from some IHC staining.

It would be useful to indicate the lineage of the virus used for inoculation. A virus name is given, but a reference to where its genome sequence is available seems to be missing, nor is a lineage designated.

Reviewer #2 (Remarks to the Author):

This manuscript reports an animal model of SARS-CoV-2-induced alveolar damage followed by regeneration that includes ADI cells, M2 macrophages, and fibrosis, which phenocopies dysregulated alveolar regeneration in COVID-19 patients. Particularly given the absence of faithful mouse models, this hamster model will be valuable to researchers in the future for investigation of disease mechanism and drug testing. The studies are performed well, the data is convincing, and the manuscript is well written. The major concern is the limited novelty.

Major Comments:

1. As the authors state in line 80, ADI cells have been observed in many models of lung injury and in human lung injury due to many causes, including COVID-19. They are found in all cases of lung injury, so the finding that this hamster model of COVID has ADI cells is not surprising.
2. The AT2-ADI-AT1 trajectory has also been shown in humans and animal models of lung injury.
3. It is also not surprising that fibrosis develops, as it does in many animal models of severe lung injury.
4. To demonstrate cell fate of CK14 cells or type 2 cells requires lineage tracing.
5. It is unclear why single cell RNAseq was done, since the authors had already identified ADI cells by IHC and could have easily and more cheaply confirmed the other pathways by IHC. Many single cell RNA seq studies have been done on many animal models of lung injury and they have all shown the same findings that are presented here.

Minor Comments:

1. Paragraph 1 is misleading. Though 49% or 23% (unclear what they are saying the difference is) of patients who were hospitalized may have chronic shortness of breath, nowhere near that many patients have fibrosis, as is implied by the sentence in line 61.
2. The papers that first described the ADI state (refs 18-21) specifically demonstrated that EMT does not occur in ADI cells, so the term EMT should be removed or restated.

Reviewer #3 (Remarks to the Author):

Summary:

I focussed on reviewing aspects of the manuscript that relate to scRNA-seq.

The authors reanalyzed public data (Nouailles, G., Wyler, E., Pennitz, P. et al. Temporal omics analysis in Syrian hamsters unravel cellular effector responses to moderate COVID-19. Nat Commun 12, 4869 (2021)) which was collected in very similar scenario (same organism, Sars-CoV2 infection, and similar time interval after infection). They subselected AT1 and AT2 cells based on published markers and used these for. They proceed to compute cell identity scores and compare composition at two different time points and cell identity as well as gene program scores across subclusters.

Major comments:

- The paragraph starting at line 565 is too qualitative, please use statistics to 1) define your hypotheses / findings and 2) make significance or effect size statements about them. As is, you are describing aspects of the data that may be meaningless in isolation. This reflects in the data presentation in Fig. 6 and 7 which is only UMAPs with scores superimposed. Try focus attention by deploying visualisation tools that contrast scores in subpopulations of interest etc. Examples include:

+ 565 "At 5 dpi, the AT1 marker *Rtkn2* was expressed in a small number of cells in clusters 2 566 and 6 (Fig. 6 A)."

Please supply statistics.

+ 566 "The AT2 marker *Lamp3* was mostly expressed in many cells within a 567 separate cell population, comprised of clusters 0, 1, 3, 4 and 5."

Please supply statistics.

The same holds for the gene set analyses starting in line 601.

- 569 "Many cells did not express any one of the two genes."

scRNAseq is a measurement that is subject to strong count noise. Given the low number of cells considered here, I would recommend being careful with making statements on absolute gene expression of individual cells. It would be much safer to argue in terms of mean expression per sub cluster if you need to make this point.

- "595 Taken together, transcriptome analysis identified AT1, AT2 and ADI cells in SARS-CoV-2-infected hamsters."

The argument on the presence of ADI is based on curated gene module scoring, you would at least need statistical tests on differential expression of key ADI marker genes between subclusters within a time point to show that these actually separate. Note that the ADI score seems to correlate with the AT1 score in Fig. 6 which suggests that the distinction of what you claim to be ADI and AT1 may not be so clear in this data.

- "596 CoV-2-infected hamsters. At 5 dpi, ADI cells did not form a separate cluster, but were admixed with AT1 and AT2 cells. At 14 dpi, ADI cells were more numerous and clustered separately from AT1 and AT2 cells."

This statement cannot be included in this form: it is neither statistically backed nor, in many cases, easy to support even with further analyses. Note that UMAP does not necessarily preserve distances in the original space.

- Given that proliferation is an effect in this system, it would be helpful to compute cell cycle scores and compare those against your cell identity scores to get an intuition for confounding sources of variation.

- "629 In summary, the findings from the independent study confirmed that ADI cells are a feature of alveolar regeneration in hamsters on a transcriptome level,"

As outlined above, I do not think that this is a direct conclusion from the presented analyses.

- Reasoning based on scRNA-seq data: In general, this section is very descriptive. I would recommend the authors to re-work their line of reasoning here to specifically converge on a hypothesis in each section. As is now, the text mostly describes variation that exists in the data, which is not necessarily causally linked to their hypotheses.

Minor comments:

- Fig. 6 and 7 have very bad image quality in the merged pdf and are almost not readable, please look into that.

- 550 "First, we generated a Uniform Manifold Approximation and Projection (UMAP) clustering all cell populations detected in the datasets."

The UMAP does not cluster cells, it is an embedding. Please correct this phrasing and declare the type of clustering you are using here.

REVIEWERS COMMENTS

Reviewer #1 (Remarks to the Author):

1) *“The manuscript by Heydemann et al. provides a detailed histology-based description and characterization of alveolar regeneration following SARS-CoV-2 infection in hamsters. This appears to be the first such detailed description of this process in hamsters, has accompanying data from human autopsy specimens and has found publicly available transcriptomic data for a separate study that largely supports the observations of this study. ”*

We thank Reviewer 1 for the positive evaluation of our work. The current version of the manuscript has been revised based on Reviewer 1 valuable suggestions that will improve its scientific contents.

2) *“The only major criticisms would be that perhaps many of the findings are not unexpected. However, given that this is not previously thoroughly described, this data set appears to be novel. And PASC is characterized in humans at the >12 weeks mark after acute infection, but this study did not assess timepoints beyond 14 days post-infection, which in the hamster model is only 7 days beyond when virus can be detected. The authors do highlight this as a limitation and suggest that future work should address this. ”*

We agree with Reviewer 1 that our study does not provide timepoints that would allow a direct comparison with PASC in humans. As Reviewer 1 indicated, we indeed underlined this fact in the “study limitation” section in our discussion. Our rationale was to first thoroughly confirm that the hamster model is suitable to study lung regeneration after SARS-CoV-2 infection. This is a prerequisite for future studies including longer timepoints that will allow to evaluate whether PASC occurs in the hamster model. These studies should also ideally involve the assessment of *in vivo* lung function in order to correlate them with histopathological findings.

3) “It would appear that this work may have been part of another study. Reference to the study seems to be missing (if it is published). This would be helpful as there is no virus data included in this paper, apart from some IHC staining. “

We thank Reviewer 1 for this valuable comment. In order to provide the reader with more information about the virus, other than SARS-CoV-2 NP immunolabellings, we added the reference of the pre-print <https://europepmc.org/article/ppr/ppr429853>) which shows viral titers of the lungs of the same animals.

This work was initially submitted to different journals since December 2021 and the viral titers data were always included there, for this reason we cannot use this data here but we will just refer to the preprint

The following changes have been made:

RESULTS

-now lines 122-128 read as follows: “First, successful infection was confirmed by immunohistochemistry for SARS-CoV-2 nucleoprotein (NP) antigen in lung tissue. Viral antigen was found in alveolar and bronchial epithelia as well as in macrophages (Fig. 1 A), as described previously 32. Quantification of immunolabeled cells in whole lung sections peaked at 3 dpi, followed by a sharp decline at 6 dpi and virus clearance at 14 dpi (Fig. 1 A). No SARS-CoV-2⁺ cells were detected in mock-infected animals at any time point. More information about viral titers in the lung of the animals investigated in this study can be found here³³.”

4) “It would be useful to indicate the lineage of the virus used for inoculation. A virus name is given, but a reference to where its genome sequence is available seems to be missing, nor is a lineage designated.”

We apologize to Reviewer 1 for the lack of clarity. Now all information have been added as suggested.

The following changes have been made:

MATERIAL AND METHODS

-now lines 936-944 read as follows:

“Virus.

SARS-CoV-2/Germany/Hamburg/01/2020 is a SARS-CoV-2 ancestral strain and was isolated from a nasopharyngeal swab of a confirmed COVID-19 patient. Stock virus was produced after three serial passages in Vero E6 cells using Dulbecco’s Modified Eagle’s Medium (DMEM; Sigma) supplemented with 2 % fetal bovine serum, 1 % penicillin-streptomycin and 1 % L-glutamine at 37 °C. The infection experiment was carried out under biosafety level 3 (BSL-3) conditions at the Leibniz Institute for Experimental Virology in Hamburg, Germany. Information about the virus genome sequence are available here: European Nucleotide Archive (ENA), study: PRJEB41216, sample: ERS5312751; accession number: SAMEA7556109.”

Reviewer #2 (Remarks to the Author):

- 1) ***“This manuscript reports an animal model of SARS-CoV-2-induced alveolar damage followed by regeneration that includes ADI cells, M2 macrophages, and fibrosis, which phenocopies dysregulated alveolar regeneration in COVID-19 patients. Particularly given the absence of faithful mouse models, this hamster model will be valuable to researchers in the future for investigation of disease mechanism and drug testing. The studies are performed well, the data is convincing, and the manuscript is well written. The major concern is the limited novelty. “***

We thank Reviewer 2 for the appreciation and for seeing in our work potential value for future scientific contributions in the field. Moreover, we would like to point out that this is the first detailed and thorough description of alveolar regeneration following SARS-CoV-2 infection in hamsters, which will have a potential impact on future studies in the field, as already stated by the kind comment of Reviewer 2. We addressed Reviewer 2 comments and concerns and we adapted the manuscript according to the valuable suggestions, which has improved the scientific value of the work. We hope that the current version of the manuscript is suitable for publication in its current form.

- 2) ***“As the authors state in line 80, ADI cells have been observed in many models of lung injury and in human lung injury due to many causes, including COVID-19. They are found in all cases of lung injury, so the finding that this hamster model of COVID has ADI cells is not surprising. The AT2-ADI-AT1 trajectory has also been shown in humans and animal models of lung injury. It is also not surprising that fibrosis develops, as it does in many animal models of severe lung injury. “***

We thank Reviewer 2 for these insightful comments. The authors agree that ADI cells are found in many cases of lung injury and that the AT2-ADI-AT1 trajectory has been described in murine models and humans. However, a detailed characterization of ADI cells has so far only been performed in mouse models of severe lung injury and pulmonary fibrosis (Jiang et al, 2020; *Am J Respir Crit Care Med*; DOI: 10.1164/rccm.201909-1726LE; Strunz et al, 2020; *Nat Commun*; DOI: doi:10.1038/s41467-020-17358-3; Riemondy et al, 2019; *JCI Insight*; DOI:10.1172/jci.insight.123637; Kobayashi et al, 2020; *Nat. Cell Biol.* DOI:10.1038/s41556-020-0542-8) and it has not been demonstrated that the process is the same in other species, including hamsters. To our knowledge, the work presented here is the first characterization of ADI in a non-murine

laboratory animal model on a morphologic and transcriptomic level. While we see that the AT2-ADI trajectory and fibrosis are comparable to mice, we also detected some differences, which show that the hamster could be more suitable for modeling the human situation. For instance, it is reported that airway-derived progenitor cells are mobilized to aid alveolar regeneration in human COVID-19 featuring severe alveolar damage (Zhao et al, 2020; *Cell Prolif*; DOI:10.1111/cpr.12931). In a recent study performed in Balb/c mice infected with mouse-adapted SARS-CoV-2 (MA-10), no contribution of airway progenitors to alveolar regeneration has been observed (Dinnon et al, 2022; *Sci Transl Med*; DOI:10.1126/scitranslmed.abo5070). Here, we show that this feature is present in hamsters, indicating that the model phenocopies this aspect of lung regeneration observed in humans more closely than mice. As Reviewer 2 has pointed out, there are no faithful mouse models of COVID-19, while the hamster model has been proven to recapitulate moderate to severe COVID-19 (Sia et al, 2020; *Nature*; DOI:10.1038/s41586-020-2342-5; Muñoz-Fontela et al, 2020; *Nature*; DOI:10.1038/s41586-020-2787-6) and is widely used to evaluate therapeutic and prophylactic measures. A comprehensive assessment of therapeutic efficacy should also include the evaluation of lung regeneration after SARS-CoV-2 induced damage. The presented work is providing a hamster-specific toolkit for this purpose.

3) *“To demonstrate cell fate of CK14 cells or type 2 cells requires lineage tracing.”*

We thank Reviewer 2 for their valuable comment. We agree that definite statements about cell fate can only be made with lineage tracing, which was unfortunately not part of the original study design. This limitation was mentioned in the submitted version of the manuscript and has been rephrased in the revised version to emphasize the need for additional studies to draw final conclusions.

The following changes have been made:

DISCUSSION

-now lines 891-895 read as follows: “Third, the conclusions regarding cell origins of ADI as well as cell fate of CK14⁺ basal cells and AT2 cells in this work are based on double-labelling and co-expression of genes interpreted in the context of published data and therefore have to be interpreted as preliminary observations. Additional studies involving lineage-tracing are required to confirm our interpretation and irrefutably prove cell trajectories.”

- 4) ***“It is unclear why single cell RNAseq was done, since the authors had already identified ADI cells by IHC and could have easily and more cheaply confirmed the other pathways by IHC. Many single cell RNA seq studies have been done on many animal models of lung injury and they have all shown the same findings that are presented here. ”***

We apologize to Reviewer 2 for the lack of clarity. We did not perform scRNAseq from tissues collected during the described animal experiment, but re-analyzed a publicly available scRNAseq dataset from a published experiment with a similar study design (Nouailles et al. 2021; *Nature Communications*; DOI: 10.1038/s41467-021-25030-7; dataset: GSE162208). A GEO search for suitable databases yielded only this one study that contained a large enough sample size of hamster lungs after infection with SARS-CoV-2 and featured data from two different timepoints covering the acute disease as well as the recovery period, similar to our study design. The selection process has now been specified in the revised manuscript. We feel that matching data from an independent experiment strengthens and validates the conclusions drawn from morphologic observations. Moreover, analyzing transcriptome data allows an investigation of many targets at the same time, which is not easy to recreate by immunostaining, particularly given the limited availability of antibodies reactive with hamster proteins. By analyzing single cell data, we were able to create hamster specific signature gene lists for alveolar cell populations, including AT1, AT2 and ADI cells, which was not available for this species before and will aid the analysis of these cell populations in future studies in the field. We agree with Reviewer 2 that the pathway analysis yielded results that can be aligned with previous findings. However, we would like to point out that this kind of analysis focusing on the transcriptome of cells involved in alveolar regeneration has not been performed in the hamster species before. We feel that the concordance of our findings with those obtained in other lung injury murine models and human COVID-19 patients strengthens the conclusion that the hamster is a good model for studying alveolar regeneration post SARS-CoV-2 infection and could fill some gaps that other models cannot.

Finally, we would like to point out that the re-analysis of the scRNAseq study is not a major part of our manuscript, but rather a complementary element of the work. Nevertheless, because of the many comments from Reviewer 3, for the revised version of the manuscript we added many additional results in the supplementary information without emphasizing this part more in the manuscript (see also responses to Reviewer 3).

The following changes have been made:

MATERIALS AND METHODS

-now lines 1049-1055 read as follows: “The dataset was selected based on a GEO search using the terms “hamster”, “SARS”, “lung”, and “single cell”. The results were filtered by “organism” (*Mesocricetus auratus*) and “study type” (Expression profiling by high throughput sequencing). The search yielded 11 studies, 8 of which included data of lung tissues from SARS-CoV-2 infected golden Syrian hamsters. From these, only one study (Nouailles et al. GSE162208) contained datasets from two different timepoints, including a timepoint after resolution of acute infection.”

- 5) ***“Paragraph 1 is misleading. Though 49% or 23% (unclear what they are saying the difference is) of patients who were hospitalized may have chronic shortness of breath, no where near that many patients have fibrosis, as is implied by the sentence in line 61.”***

We apologize for the lack of clarity and agree with Reviewer 2 that the phrasing might be misleading. To clarify that not all people suffering from PASC symptoms have evidence of lung fibrosis, the paragraph was rephrased.

The following changes have been made:

INTRODUCTION

-now lines 57-69 read as follows: “PASC occurs in 3-11.7% of infected individuals and is characterized by symptoms such as fatigue, headache, cognitive dysfunction, altered smell and taste, shortness of breath, and dyspnea, occurring >12 weeks after acute virus infection^{4,5}. Of note, among patients with severe disease requiring hospitalization, respiratory symptoms are reported with a much higher frequency. For instance, shortness of breath occurs in up to 49% and dyspnea in up to 23.3% of patients for 8-10 months after acute disease^{6,7}. The pathomorphological correlates and mechanisms responsible for respiratory PASC are still not fully understood. It is known that lung fibrosis is a potential sequel to COVID-19 pneumonia^{8,9}. CT scans 4 months after infection show that fibrosis develops in up to 21% of patients who survived severe COVID-19¹⁰. However, fibrosis cannot account for all respiratory PASC symptoms. An additional suggested mechanism is a prolonged impairment of gas exchange capacity due to an incomplete or protracted regeneration of alveoli^{8,11}.”

- 6) ***“The papers that first described the ADI state (refs 18-21) specifically demonstrated that EMT does not occur in ADI cells, so the term EMT should be removed or restated. ”***

We thank Reviewer 2 for their valuable and correct observation. In order to clarify this aspect, the sentence has been modified in the revised version of the manuscript.

The following changes have been made:

DISCUSSION

-now lines 831-840 read as follows: “Of note, it has been reported that ADI cells can display high scores for EMT genes¹⁹ and the results from our transcriptome analysis showed that cells in clusters with AT1/ADI gene expression also showed high EMT pathway module scores at 5 and 14 dpi in SARS-CoV-2 infected hamsters. However, an EMT process with conversion of ADI cells into fibroblasts or other mesenchymal cells could not be demonstrated in murine models of lung injury¹⁸⁻²¹. Thus, it has been postulated that upregulation of genes typically involved in this process is necessary for the morphological changes that ADI cells undergo completing the AT2-ADI-AT1 trajectory (i.e. flattening and spread), rather than entering a mesenchymal-like cell state¹⁹.”

Reviewer #3 (Remarks to the Author):

- 1) ***“I focussed on reviewing aspects of the manuscript that relate to scRNA-seq. The authors reanalyzed public data (Nouailles, G., Wyler, E., Pennitz, P. et al. Temporal omics analysis in Syrian hamsters unravel cellular effector responses to moderate COVID-19. Nat Commun 12, 4869 (2021)) which was collected in very similar scenario (same organism, Sars-CoV2 infection, and similar time interval after infection). They subselected AT1 and AT2 cells based on published markers and used these for. They procede to compute cell identity scores and compare composition at two different time points and cell identity as well as gene program scores across subclusters.”***

We thank Reviewer 3 for the time and the effort to review our manuscript. The authors highly appreciate the very helpful comments of Reviewer 3. Our scRNAseq study was based on published data for which we have only limited information and many details are missing. We acknowledge the reviewer’s comments that we might have over interpreted and not correctly described our results at several manuscript sections (see further comments of Reviewer 3 below). For this reason, we have now adapted and tuned down the scRNAseq data analysis paragraph and the respective subheading. In addition, we now rephrased our statements about expression in cells or clusters when referring to results from module scores. Instead now we use the term ‘high module scores’. Moreover, as suggested by Reviewer 3, we now provide a statistical analysis of *Rtkn2* and *Lamp3* gene expression with an additional visualization of the results (bar plots), which display the expression differences among clusters more distinctively, and we added information on significance (Fig. 6). The module score analysis is explained more clearly, and more data are provided that rationalize the choice of genes used in the module scores for AT1, AT2, ADI, club and ciliated cells. The selection of these genes was based on a multistep process. First, we determined differentially expressed genes (DEGs) for all clusters. These results, including average expression levels per cluster, multiple-testing adjusted p-values and fold changes, are now included in the supplementary material (Supplementary Table 1 and 2). In the second step, we selected the top 10 DEGs in each cluster and determined the cell type in which they are typically expressed. To do this we used literature from mouse models of lung regeneration and COVID-19 that also employed scRNASeq analysis. We then created a hamster-specific candidate marker gene list for all the cell types mentioned above. In a final step, the specificity and distribution of expression among clusters was determined and genes with ubiquitous expression were removed from the lists. The final gene list as well as the feature and ridge plots

for all these genes are now provided in the supplementary material (Supplementary Fig. 8-27). Therefore, in addition to displaying module scores for gene sets, we now also provide information on all single genes and we added a display that contrasts the differences between clusters better, as requested by Reviewer 3. Since the focus of our analysis was the detection of ADI cells, we provide statistical analysis of the expression of all individual ADI cell genes used in the module scores, to back up our conclusions on cell identity (Supplementary Fig. 28-29; Supplementary Table 4, 5, and 6). Lastly, we clearly state when we draw a hypothesis from our data throughout the section, and we added a paragraph in the discussion stating the limitations of our analysis. We hope that the current version of the manuscript fulfills all requirements requested by Reviewer 3.

The following changes have been made:

RESULTS (scRNAseq data analysis paragraph subheading)

-now lines 553-554 read as follows: **“5. Single-cell transcriptome analysis supports ADI cell persistence following SARS-CoV-2 induced DAD in hamsters.”**

In order to adjust the manuscript regarding scRNAseq new analysis, substantial changes have been done to the results section (present in the track-changes manuscript version), here we report only the clean version.

RESULTS

-now lines 555-696 read as follows:

“As described above, we demonstrated that ADI cells with features previously described in mouse models of lung regeneration as well as in COVID-19 patients are participating in alveolar regeneration following SARS-CoV-2 infection of hamsters. To support this observation we wanted to detect ADI also on a transcriptome level, using data from an independent experiment. In order to do so, we re-analyzed a previously published single-cell RNASeq gene expression dataset (GSE162208) from lungs of SARS-CoV-2 infected Syrian golden hamsters⁴³. The experiment was performed with a study design similar to this investigation. We focused our analysis on data from SARS-CoV-2-infected animals sacrificed at 5 and 14 dpi. First, we generated a Uniform Manifold Approximation and Projection (UMAP) embedding using the Seurat package (functions: RunPCA, FindNeighbors, FindClusters⁴⁴⁻⁴⁹) for clustering to visualize and identify cell populations based on similar and known/presumed expression

profiles. We then identified alveolar cells based on the expression of AT1 and AT2 markers (*Rtkn2* and *Lamp3*, respectively), as described in the original publication (Fig. 6 A-F)⁴³. These cells were then re-clustered resulting in 7 and 11 clusters at 5 and 14 dpi, respectively (Fig. 6 B-G).

At 5 dpi, significantly high mean expression levels of *Rtkn2* were seen in clusters 2 and 6 (Fig. 6 C). Significantly high mean expression levels of *Lamp3* were detected in clusters 0, 3, and 5 (Fig. 6 D). Clusters 1 and 4 did not show significantly high mean expression levels for any of the two markers. Based on this, clusters 2 and 6 were considered AT1, clusters 0, 3, and 5 AT2 and clusters 1 and 4 unknown. Interestingly, clusters 2, 5 and 6 showed a variably high double expression of *Rtkn2* and *Lamp3* (Fig. 6 E).

At 14 dpi, significantly high mean expression levels of *Rtkn2* were seen in clusters 2, 5 and 7 (Fig. 6 H). Significantly high mean expression levels of *Lamp3* were detected in clusters 4, 6, and 8 (Fig. 6 I). Clusters 0, 1, 3, 9, and 10 did not show a significantly high mean expression level for neither *Rtkn2* nor *Lamp3*. Based on this, clusters 2, 5, and 7 were considered AT1, clusters 4, 6, and 8 AT2 and clusters 0, 1, 3, 9, and 10 unknown. As for 5 dpi, a variably high double expression of *Rtkn2* and *Lamp3* was detected in multiple clusters (0, 1, 2, 4, 5, 7, 9) as shown in Fig. 6 J. Cluster 10 did not show any expression of the investigated genes.

As mentioned before, ADI cells are known to arise from AT2, enter this transient distinct ADI cell state and then transdifferentiate into AT1. For this reason “early” ADI cells might show an overlap in gene and protein expression with AT2 cells, as demonstrated by immunofluorescence double labeling in this work. It has been reported that the full ADI cell state is distinct and ADI cells could form a separate cluster independent from AT2 or AT1 clusters¹⁹. “Late” ADI cells might show an overlapping gene and protein expression with AT1 cells. Based on this, our hypothesis was that different stages of ADI cells could be located in clusters of the scRNAseq analysis showing a double expression of AT1 and AT2 markers or in those clusters that were not clearly identifiable as AT2 or AT1.

Therefore, we analyzed gene expression profiles of all clusters in order to determine the distribution of cells expressing ADI genes. First, we identified the top 10 differentially expressed genes (DEGs) in each cluster (Supplementary Table 1 and 2) and compared the sets of DEGs with gene signatures described in mouse models of lung regeneration^{19,20} as well as COVID-19 patients⁹. Within the DEGs, we detected genes typically expressed by AT1, AT2, ADI cells, club cells or ciliated cells in mice and/or humans. We then generated a list of candidate marker genes for these cell types in the hamster. Next, we evaluated the expression of these candidate markers within the clusters and removed genes with low specificity from the lists. The final, hamster-specific marker gene lists

are given in Supplementary Table 3 and the details of the expression distribution within clusters are shown as feature plots and ridge plots in Supplementary Fig. 8-27. We then applied the module score algorithm (function `AddModuleScore` from Seurat package⁴⁴⁻⁴⁹) with these sets of marker genes on the clusters detected at 5 and 14 dpi.

At 5 dpi, high module scores for AT1 cell markers were seen in cluster 6 and partly in cluster 2, in line with the distribution of *Rtkn2* expression (Fig. 7 A). High module scores for AT2 cell markers were found in clusters 0, 1, 3, 4 and 5 (Fig. 7 B), partially in line with *Lamp3* distribution. High module scores for ADI cell markers were detected throughout clusters 2 and 6 and partly in cluster 5, which were the clusters that showed a double expression of *Rtkn2* and *Lamp3* (Fig. 7 C). Interestingly, clusters showing high module scores for AT2 cell markers also partly showed high module scores for club cell markers (cluster 1, 3 and 5, Fig. 7 D) and ciliated cell markers (Cluster 1, Fig. 7 E). Interestingly, clusters 1 and 4 that were previously noted as unknown, showed high module scores for AT2, club and ciliated cell markers or only AT2 cell markers, respectively.

At 14 dpi, high module scores for AT1 cell markers were seen in clusters 2, 5, and partly in cluster 7, in line with *Rtkn2* expression (Fig. 7 F). High module scores for AT2 cell markers were found in clusters 1, 4, 6 and 8 (Fig. 7 G), partially in line with *Lamp3* distribution. High module scores for ADI cell markers were detected in clusters 2, 5, 7, 9, and partly in clusters 0 and 3 (Fig. 7 H). Clusters showing high module scores for AT2 cell markers also partly showed high module scores for club cell markers (cluster 1, 4, 6, and 8 Fig. 7 I). Interestingly, cluster 1 that was previously noted as unknown, showed high module scores for AT2 and club cell markers. Other unknown clusters like cluster 0, 3, and 9, showed high module scores for ADI cell markers. In addition, clusters 2, 5, 7 that previously showed a double expression of *Rtkn2* and *Lamp3*, showed also high module scores for ADI cell markers. Cluster 10, previously unknown, separated completely from the other populations and showed a high score for ciliated cell markers (Fig. 7 J).

Since our aim was to identify ADI cells, we also investigated the mean expression levels of all individual genes in the ADI cell marker list in each cluster (Supplementary Table 3) to support the module score analysis results. The bar plots with statistical analysis and details on gene expression within clusters are given in Supplementary Fig. 28-29 and Supplementary Table 4, 5, and 6. At 5 dpi, clusters 2 and 6 displayed significantly high mean expression levels for at least 8 ADI genes (cluster 2: 8/28 genes; cluster 6: 19/28 genes; Supplementary Fig. 28 and Supplementary Table 4 and 5). Cluster 5 did not show significantly high mean expression levels for any of the investigated ADI genes. These results partly

confirmed the module score results and suggest that both AT1 and ADI cells belong to clusters 2 and 6.

At 14 dpi, clusters 0, 2, 3, 5, 7, and 9 displayed significantly high mean expression levels for at least 3 ADI genes (cluster 0: 6/28 genes; cluster 2: 10/28 genes; cluster 3: 3/28 genes; cluster 5: 9/28 genes; cluster 7: 19/28 genes; cluster 9: 9/28 genes; Supplementary Fig. 29 and Supplementary Table 4 and 6). These results confirmed the module score results and suggest that both AT1 and ADI cells belong to clusters 2, 5, and 6 as well as ADI cells belong to clusters 0, 3, and 9.

Taken together, transcriptome analysis identified clusters showing high module scores for AT1, AT2, ADI, club and ciliated cell marker gene sets. Additionally, we identified clusters with a significantly higher expression levels of ADI marker genes compared to other clusters. We interpret these differences in expression levels as absence or presence of the respective cell populations. Based on this, we assigned one or more presumable cell identities for each cluster (Fig. 8 A-B). We postulate that ADI cells showed an overlap of gene expression with AT1 cells, particularly at 5 dpi. At 14 dpi, a more distinct ADI cell population could be detected, which separated from the AT1 and AT2 clusters.

Next, we wanted to investigate if cell clusters with a high expression of ADI genes showed high module scores for pathways involved in lung regeneration. For this, we performed module score analysis with selected hallmark gene lists (<http://www.gsea-msigdb.org/gsea/msigdb/index.jsp>): *p53 pathway*, *DNA repair*, *TGF beta signaling*, *notch signaling*, *wnt beta catenin signaling*, *epithelial mesenchymal transition (EMT)*, *angiogenesis*, and *G2M checkpoint*. As described above, ADI cells in mice and humans express Tp53 and other markers of cell cycle arrest and DNA repair. The transcriptome data showed that two clusters with high ADI/AT1 module scores showed high positive scores for *p53 pathway* genes at 14 dpi (Fig. 8 C-D). At 5 dpi, almost all clusters showed positive scores for *DNA repair* genes, with the highest scores observed in cells within clusters with high AT1/ADI module scores (Fig. 8 E). At 14 dpi, mainly clusters with high AT1/ADI and ADI module scores displayed positive scores for *DNA repair* (Fig. 8 F). The AT2-ADI-AT1 trajectory is regulated by different signaling pathways, including TGF beta -, notch - and wnt beta catenin signaling and involves the EMT process^{19,44}. At 5 dpi, a cluster with high AT2/club cell module scores and a cluster with high AT1/ADI cell module scores, partly showed high positive scores for *TGF beta signaling*. At 14 dpi, high positive scores for this gene set were partly detected in one cluster with high scores for AT1/ADI (Fig. 8 G-H). A cluster with high AT2/club cells module scores revealed a high positive score for *notch signaling* hallmark genes at 5 dpi, whereas variably positive scores were distributed among clusters with high AT2/club cell, ADI and AT1/ADI cell module scores at 14 dpi (Fig. 8 I-J).

A cluster with high AT1/ADI module scores partly revealed high positive scores for *wnt beta catenin signaling* hallmark genes at 5 dpi (Fig. 8 K). At 14 dpi, mainly clusters with high AT1/ADI and ADI cell module scores showed positive scores for *wnt beta catenin signaling* hallmark genes (Fig. 8 L). In addition, clusters with high AT2/club cell module scores also partly showed high scores for *wnt beta catenin signaling* hallmark genes at 14 dpi (Fig. 8 L). High positive scores for *EMT* hallmark genes were mainly found in clusters with high AT1/ADI module scores and partly within a cluster with high AT2/club cell module scores at 5 dpi (Fig. 8 M). At 14 dpi, a cluster with high AT1/ADI module scores showed positive scores for *EMT* hallmark genes (Fig. 8 N). Subsequently, we investigated the expression of genes involved in angiogenesis, since this process is upregulated in late phases of DAD, in the context of fibrosis⁴⁵. Clusters with high AT1/ADI module scores at 5 dpi and clusters with high AT1/ADI or ADI module scores at 14 dpi revealed high positive scores for *angiogenesis* hallmark genes (Fig. 8 O-P). Finally, we wanted to investigate which cell clusters show high scores for *G2M checkpoint* hallmark genes as a readout of cell proliferation. The highest scores for this gene set were observed in clusters with high AT2/club cell and AT1/ADI cell module scores at 14 dpi (Fig. 8 Q-R).”

DISCUSSION

In order to tone down the manuscript regarding scRNAseq data and to adjust it to the new results, several sentences have been removed from the discussion (present in the track-changes manuscript version), here we report only the clean version.

-now lines 772-775 read as follows:”CK8⁺ADI cells in SARS-CoV-2 infected hamsters frequently expressed nuclear TP53 protein. Transcriptome data also showed that some clusters with high ADI gene expression displayed high module scores for *p53 pathway* and for *DNA repair* hallmark genes.”

-now lines 795-798 read as follows: ”We showed that, from 5 to 14 dpi, an increasing number of clusters with high ADI gene expression showed high scores for *Wnt/ β -catenin* and *notch signaling* hallmark genes, while module scores for *TGF- β signaling pathway* were less prominent and detected only in one cluster”

-now lines 831-840 read as follows: “ Of note, it has been reported that ADI cells can display high scores for EMT genes²⁰ and the results from our transcriptome analysis showed that cells in clusters with AT1/ADI gene expression also showed high EMT pathway module scores at 5 and 14 dpi in SARS-CoV-2 infected hamsters. However, an EMT process with conversion of ADI cells into fibroblasts or other mesenchymal cells could not been demonstrated in murine models of

lung injury¹⁹⁻²². Thus, it has been postulated that upregulation of genes typically involved in this process is necessary for the morphological changes that ADI cells undergo completing the AT2-ADI-AT1 trajectory (i.e. flattening and spread), rather than entering a mesenchymal-like cell state²⁰.”

-now lines 841-845 read as follows: “Interestingly, transcriptome data revealed that clusters with high AT1/ADI and ADI gene expression showed high positive scores for *angiogenesis* hallmark genes at 14 dpi, suggesting that ADI cells in hamsters might contribute also to a pro-angiogenetic microenvironment, promoting vascular changes during lung fibrosis similar to COVID-19 patients”

-now lines 894-900 read as follows:” Fourth, the conclusions from transcriptome analysis require confirmation in future studies. It is possible that expression of certain genes may not have been detected because of limited cell numbers or depth of sequencing. It is thus possible that cells with lower expression levels or module scores are also present in other clusters. In addition, future studies including evaluation of protein expression of genes involved in the analyzed pathways are needed also to confirm our interpretations. ”

MATERIAL AND METHODS

-now lines 1046-1080 read as follows:

“Single-cell RNAseq.

Single-cell RNASeq data from lungs of SARS-CoV-2 infected hamsters was obtained from a publicly available dataset⁴³. The dataset was selected based on a GEO search using the terms “hamster”, “SARS”, “lung”, and “single cell”. The results were filtered by organism (*Mesocricetus auratus*) and study type (expression profiling by high throughput sequencing). The search yielded 11 studies, 8 of which included data of lung tissues from SARS-CoV-2 infected golden Syrian hamsters. From these, only one study (Nouailles et al. GSE162208) contained datasets from two different timepoints, including a timepoint after resolution of acute infection. Data were analyzed using the R software package (version 3.6.0)⁷⁰. Expression data were downloaded from GEO (<https://www.ncbi.nlm.nih.gov/geo/>, GSE162208) and Seurat objects (version Seurat_3.2.0,⁷¹⁻⁷⁴ were generated from downloaded h5 files by combining replicate samples from lungs at 5 and 14 dpi. Pre-processing of data was performed by applying several Seurat functions: subset (subset = nFeature_RNA > 200 & nFeature_RNA < 2500 & percent.mt < 5), NormalizeData (data), FindVariableFeatures (data, selection.method = "vst", nfeatures = 2000), ScaleData(data, features = all.genes), and RunPCA(data, features = VariableFeatures (object = data)). Uniform Manifold Approximation and Projection (UMAP) were performed after clustering with Seurat functions

FindNeighbors(pbmc, dims = 1:10) and FindClusters(pbmc, resolution = 0.5). AT1 and AT2 cell cluster were then identified by using the marker genes *Rtkn2* (AT1) and *Lamp3* (AT2), respectively, from the original publication ⁴³. These clusters were selected, then pre-processed and re-clustered as described above. We then collected more candidate marker genes for AT1, AT2, ADI cells and additional cell populations in these clusters by applying functions FindAllMarkers (pbmc, only.pos = TRUE, min.pct = 0.25, logfc.threshold = 0.25) and selecting the top 10 significant markers genes per cluster. We further identified additional candidate markers from ^{9,19,43}. We evaluated the specificity of these candidate markers by visualizing them with the functions FeaturePlot, RidgePlot and AddModuleScore of the Seurat package. The list of final maintained marker genes is presented in supplementary table 1. The function AddModuleScore of the Seurat package was used to visualize high module scores of the different cell populations and hallmark genes from the GSEA database (<http://www.gsea-msigdb.org/gsea/msigdb> ⁷⁵). Means in expression levels were calculated using function AverageExpression of the Seurat package, differences in log-fold change and corresponding multiple testing adjusted p-values were determined using function FindAllMarkers of the Seurat package ⁴⁴⁻⁴⁹.

2) ***“ The paragraph starting at line 565 is too qualitative, please use statistics to 1) define your hypotheses / findings and 2) make significance or effect size statements about them. As is, you are describing aspects of the data that may be meaningless in isolation. This reflects in the data presentation in Fig. 6 and 7 which is only UMAPs with scores superimposed. Try focus attention by deploying visualisation tools that contrast scores in subpopulations of interest etc.***

Examples include:

+ 565 "At 5 dpi, the AT1 marker *Rtkn2* was expressed in a small number of cells in clusters 2 566 and 6 (Fig. 6 A)."--- Please supply statistics.

+ 566 "The AT2 marker *Lamp3* was mostly expressed in many cells within a 567 separate cell population, comprised of clusters 0, 1, 3, 4 and 5."--- Please supply statistics.

We thank Reviewer 3 for their valuable comments that will improve the scientific content of our manuscript. In order to address this request, we changed the manuscript, provided new figures including a statistical analysis of *Rtkn2*, *Lamp3*, and ADI cell gene expression as outlined in the comment #1.

The following changes have been made:

RESULTS

-now lines 571-585 read as follows: “At 5 dpi, significantly high mean expression levels of *Rtkn2* were seen in clusters 2 and 6 (Fig. 6 C). Significantly high mean expression levels of *Lamp3* were detected in clusters 0, 3, and 5 (Fig. 6 D). Clusters 1 and 4 did not show significantly high mean expression levels for any of the two markers. Based on this, clusters 2 and 6 were considered AT1, clusters 0, 3, and 5 AT2 and clusters 1 and 4 unknown. Interestingly, clusters 2, 5 and 6 showed a variably high double expression of *Rtkn2* and *Lamp3* (Fig. 6 E).

At 14 dpi, significantly high mean expression levels of *Rtkn2* were seen in clusters 2, 5 and 7 (Fig. 6 H). Significantly high mean expression levels of *Lamp3* were detected in clusters 4, 6, and 8 (Fig. 6 I). Clusters 0, 1, 3, 9, and 10 did not show a significantly high mean expression level for neither *Rtkn2* nor *Lamp3*. Based on this, clusters 2, 5, and 7 were considered AT1, clusters 4, 6, and 8 AT2 and clusters 0, 1, 3, 9, and 10 unknown. As for 5 dpi, a variably high double expression of *Rtkn2* and *Lamp3* was detected in multiple clusters (0, 1, 2, 4, 5, 7, 9) as shown in Fig. 6 J. Cluster 10 did not show any expression of the investigated genes.”

-now lines 634-649 read as follows: “Since our aim was to identify ADI cells, we also investigated the mean expression levels of all individual genes in the ADI cell marker list in each cluster (Supplementary Table 3) to support the module score analysis results. The bar plots with statistical analysis and the genes expression distribution within clusters are given in Supplementary Fig. 28-29 and Supplementary Table 4, 5, and 6. At 5 dpi, clusters 2 and 6 displayed significantly high mean expression levels for at least 8 ADI genes (cluster 2: 8/28 genes; cluster 6: 19/28 genes; Supplementary Fig. 28 and Supplementary Table 4, and 5). Cluster 5 did not show significantly high mean expression levels for any of the investigated ADI genes. These results partly confirmed the module score results and suggest that both AT1 and ADI cells belong to clusters 2 and 6.

At 14 dpi, clusters 0, 2, 3, 5, 7, and 9 displayed significantly high mean expression levels for at least 3 ADI genes (cluster 0: 6/28 genes; cluster 2: 10/28 genes; cluster 3: 3/28 genes; cluster 5: 9/28 genes; cluster 7: 19/28 genes; cluster 9: 9/28 genes; Supplementary Fig. 29 and Supplementary Table 4 and 6). These results confirmed the module score results and suggest that both AT1 and ADI cells belong to clusters 2, 5, and 6 as well as ADI cells belong to clusters 0, 3, and 9.”

3) *The same holds for the gene set analyses starting in line 601.” Next, we wanted to investigate the expression of genes belonging to pathways 601 involved in lung regeneration and we performed module score analysis with hallmark 602 gene lists”*

We thank Reviewer 3 for the comment. The authors feel that, since hallmark gene lists are quite large (32 to 200 genes), it would be difficult to display them by individual genes in feature plots and ridge plots or bar plots per every single

pathway. In addition, to our knowledge, using gene sets is more robust than individual gene expression as the former relies on multiple genes and is less susceptible to the excess zero counts common in scRNA-seq (A Guide to Analyzing Single-cell Datasets; John F. Ouyang, 2023, chapter 3.3.4. “calculating module scores”; <https://ouyanglab.com/singlecell/clust.html#sec:gmodu>). Therefore, we chose to apply the module scores, which are based on analysis involving the entire gene list, which we feel is a more comprehensive approach. Moreover, we have adapted the manuscript text and re-phrased it in order to refer to high module scores instead of differential gene expression. In addition, we toned down our conclusions and included study limitations.

The following changes have been made:

RESULTS

-now lines 659-696 read as follows: “Next, we wanted to investigate if cell clusters with a high expression of ADI genes showed high module scores for pathways involved in lung regeneration. For this, we performed module score analysis with selected hallmark gene lists (<http://www.gsea-msigdb.org/gsea/msigdb/index.jsp>): *p53 pathway*, *DNA repair*, *TGF beta signaling*, *notch signaling*, *wnt beta catenin signaling*, *epithelial mesenchymal transition (EMT)*, *angiogenesis*, and *G2M checkpoint*. As described above, ADI cells in mice and humans express Tp53 and other markers of cell cycle arrest and DNA repair. The transcriptome data showed that two clusters with high ADI/AT1 module scores showed high positive scores for *p53 pathway* genes at 14 dpi (Fig. 8 C-D). At 5 dpi, almost all clusters showed positive scores for *DNA repair* genes, with the highest scores observed in cells within clusters with high AT1/ADI module scores (Fig. 8 E). At 14 dpi, mainly clusters with high AT1/ADI and ADI module scores displayed positive scores for *DNA repair* (Fig. 8 F). The AT2-ADI-AT1 trajectory is regulated by different signaling pathways, including TGF beta -, notch - and wnt beta catenin signaling and involves the EMT process^{19,44}. At 5 dpi, a cluster with high AT2/club cell module scores and a cluster with high AT1/ADI cell module scores, partly showed high positive scores for *TGF beta signaling*. At 14 dpi, high positive scores for this gene set were partly detected in one cluster with high scores for AT1/ADI (Fig. 8 G-H). A cluster with high AT2/club cells module scores revealed a high positive score for *notch signaling* hallmark genes at 5 dpi, whereas variably positive scores were distributed among clusters with high AT2/club cell, ADI and AT1/ADI cell module scores at 14 dpi (Fig. 8 I-J). A cluster with high AT1/ADI module scores partly revealed high positive scores for *wnt beta catenin signaling* hallmark genes at 5 dpi (Fig. 8 K). At 14 dpi, mainly

clusters with high AT1/ADI and ADI cell module scores showed positive scores for *wnt beta catenin signaling* hallmark genes (Fig. 8 L). In addition, clusters with high AT2/club cell module scores also partly showed high scores for *wnt beta catenin signaling* hallmark genes at 14 dpi (Fig. 8 L). High positive scores for *EMT* hallmark genes were mainly found in clusters with high AT1/ADI module scores and partly within a cluster with high AT2/club cell module scores at 5 dpi (Fig. 8 M). At 14 dpi, a cluster with high AT1/ADI module scores showed positive scores for *EMT* hallmark genes (Fig. 8 N). Subsequently, we investigated the expression of genes involved in angiogenesis, since this process is upregulated in late phases of DAD, in the context of fibrosis⁴⁵. Clusters with high AT1/ADI module scores at 5 dpi and clusters with high AT1/ADI or ADI module scores at 14 dpi revealed high positive scores for *angiogenesis* hallmark genes (Fig. 8 O-P). Finally, we wanted to investigate which cell clusters show high scores for *G2M checkpoint* hallmark genes as a readout of cell proliferation. The highest scores for this gene set were observed in clusters with high AT2/club cell and AT1/ADI cell module scores at 14 dpi (Fig. 8 Q-R)."

DISCUSSION

-now lines 772-775 read as follows: "CK8⁺ADI cells in SARS-CoV-2 infected hamsters frequently expressed nuclear TP53 protein. Transcriptome data also showed that some clusters with high ADI gene expression displayed high module scores for *p53 pathway* and for *DNA repair* hallmark genes."

-now lines 795-798 read as follows: "We showed that, from 5 to 14 dpi, an increasing number of clusters with high ADI gene expression showed high scores for *Wnt/β-catenin* and *notch signaling* hallmark genes, while module scores for *TGF-β signaling pathway* were less prominent and detected only in one cluster"

-now lines 831-840 read as follows: "Of note, it has been reported that ADI cells can display high scores for EMT genes²⁰ and the results from our transcriptome analysis showed that cells in clusters with AT1/ADI gene expression also showed high EMT pathway module scores at 5 and 14 dpi in SARS-CoV-2 infected hamsters. However, an EMT process with conversion of ADI cells into fibroblasts or other mesenchymal cells could not be demonstrated in murine models of lung injury¹⁹⁻²². Thus, it has been postulated that upregulation of genes typically involved in this process is necessary for the morphological changes that ADI cells undergo completing the AT2-ADI-AT1 trajectory (i.e. flattening and spread), rather than entering a mesenchymal-like cell state²⁰."

-now lines 841-845 read as follows: “Interestingly, transcriptome data revealed that clusters with high AT1/ADI and ADI gene expression showed high positive scores for *angiogenesis* hallmark genes at 14 dpi, suggesting that ADI cells in hamsters might contribute also to a pro-angiogenetic microenvironment, promoting vascular changes during lung fibrosis similar to COVID-19 patients”

-now lines 894-900 read as follows:” Fourth, the conclusions from transcriptome analysis require confirmation in future studies. It is possible that expression of certain genes may not have been detected because of limited cell numbers or depth of sequencing. It is thus possible that cells with lower expression levels or module scores are also present in other clusters. In addition, future studies including evaluation of protein expression of genes involved in the analyzed pathways are needed also to confirm our interpretations. ”

4) ***““569 "Many cells did not express any one of the two genes." scRNAseq is a measurement that is subject to strong count noise. Given the low number of cells considered here, I would recommend being careful with making statements on absolute gene expression of individual cells. It would be much safer to argue in terms of mean expression per sub cluster if you need to make this point.”***

The authors apologize to Reviewer 3 for the lack of clarity. We agree that we should not make conclusions whether a gene is expressed in a given cell at all or not, but we can make conclusions about its module scores. For this reason we removed this statement accordingly. In addition, as suggested by Reviewer 3, we investigated the mean expression level for *Rtkn2* and *Lamp3* as well as ADI cell genes in order to make the point clear in the most critical passages of the manuscript. Additionally, we point out the limitations of our study in this respect in the discussion section.

The following changes have been made:

RESULTS

-now lines 571-585 read as follows: “At 5 dpi, significantly high mean expression levels of *Rtkn2* were seen in clusters 2 and 6 (Fig. 6 C). Significantly high mean expression levels of *Lamp3* were detected in clusters 0, 3, and 5 (Fig. 6 D). Clusters 1 and 4 did not show significantly high mean expression levels for any of the two markers. Based on this, clusters 2 and 6 were considered AT1, clusters

0, 3, and 5 AT2 and clusters 1 and 4 unknown. Interestingly, clusters 2, 5 and 6 showed a variably high double expression of *Rtkn2* and *Lamp3* (Fig. 6 E).

At 14 dpi, significantly high mean expression levels of *Rtkn2* were seen in clusters 2, 5 and 7 (Fig. 6 H). Significantly high mean expression levels of *Lamp3* were detected in clusters 4, 6, and 8 (Fig. 6 I). Clusters 0, 1, 3, 9, and 10 did not show a significantly high mean expression level for neither *Rtkn2* nor *Lamp3*. Based on this, clusters 2, 5, and 7 were considered AT1, clusters 4, 6, and 8 AT2 and clusters 0, 1, 3, 9, and 10 unknown. As for 5 dpi, a variably high double expression of *Rtkn2* and *Lamp3* was detected in multiple clusters (0, 1, 2, 4, 5, 7, 9) as shown in Fig. 6 J. Cluster 10 did not show any expression of the investigated genes.”

-now lines 634-649 read as follows: “Since our aim was to identify ADI cells, we also investigated the mean expression levels of all individual genes in the ADI cell marker list in each cluster (Supplementary Table 3) to support the module score analysis results. The bar plots with statistical analysis and the genes expression distribution within clusters are given in Supplementary Fig. 28-29 and Supplementary Table 4, 5, and 6. At 5 dpi, clusters 2 and 6 displayed significantly high mean expression levels for at least 8 ADI genes (cluster 2: 8/28 genes; cluster 6: 19/28 genes; Supplementary Fig. 28 and Supplementary Table 4 and 5). Cluster 5 did not show significantly high mean expression levels for any of the investigated ADI genes. These results partly confirmed the module score results and suggest that both AT1 and ADI cells belong to clusters 2 and 6.

At 14 dpi, clusters 0, 2, 3, 5, 7, and 9 displayed significantly high mean expression levels for at least 3 ADI genes (cluster 0: 6/28 genes; cluster 2: 10/28 genes; cluster 3: 3/28 genes; cluster 5: 9/28 genes; cluster 7: 19/28 genes; cluster 9: 9/28 genes; Supplementary Fig. 29 and Supplementary Table 4 and 6). These results confirmed the module score results and suggest that both AT1 and ADI cells belong to clusters 2, 5, and 6 as well as ADI cells belong to clusters 0, 3, and 9.”

DISCUSSION

-now lines 894-900 read as follows: “Fourth, the conclusions from transcriptome analysis require confirmation in future studies. It is possible that expression of certain genes may not have been detected because of limited cell numbers or depth of sequencing. It is thus possible that cells with lower expression levels or module scores are also present in other clusters. In addition, future studies including evaluation of protein expression of genes involved in the analyzed pathways are needed also to confirm our interpretations. ”

- 5) ***“595 Taken together, transcriptome analysis identified AT1, AT2 and ADI cells in SARS-CoV-2-infected hamsters.” The argument on the presence of ADI is based on curated gene module scoring, you would at least need statistical tests on differential expression of key ADI marker genes between subclusters within a time point to show that these actually separate. Note that the ADI score seems to correlate with the AT1 score in Fig. 6 which suggests that the distinction of what you claim to be ADI and AT1 may not be so clear in this data.”***

We thank Reviewer 3 for this valuable comment. As suggested by Reviewer 3, we now included mean expression levels with statistical tests for 28 key ADI genes at both time points (Supplementary Fig. 28-29; Supplementary Table 4). This additional analysis supports our module score results, and we believe that it is allowing us to postulate with more precision which clusters contain ADI cells (cluster 2 ad 6 at 5 dpi; cluster 0, 2, 3, 5, 7, and 9 at 14 dpi). At 5 dpi, both clusters have significantly high mean expression levels for AT1 and ADI, the same goes for clusters 2, 5, and 7 at 14 dpi. At 14 dpi, clusters 0, 3, and 9 show significantly high expression of ADI genes, but not AT1 or AT2 genes, indicating that ADI cells separate more clearly at this timepoint. As mentioned in our manuscript, ADI cells are known to arise from AT2, enter this transient distinct ADI cell state and then transdifferentiate into AT1. For this reason, “early” ADI cells might show an overlap in gene and protein expression with AT2 cells and “late” ADI cells might show an overlapping gene and protein expression with AT1 cells. Therefore, it is to be expected that the separation will not always be very clear. To highlight this point, we included an additional paragraph in the results section.

The following changes have been made:

RESULTS

-now lines 586-595 read as follows: “As mentioned before, ADI cells are known to arise from AT2, enter this transient distinct ADI cell state and then transdifferentiate into AT1. For this reason “early” ADI cells might show an overlap in gene and protein expression with AT2 cells, as demonstrated by immunofluorescence double labeling in this work. It has been reported that the full ADI cell state is distinct and ADI cells could form a separate cluster independent from AT2 or AT1 clusters ¹⁹. “Late” ADI cells might show an overlapping gene and protein expression with AT1 cells. Based on this, our hypothesis was that different stages of ADI cells could be located in clusters of

the scRNAseq analysis showing a double expression of AT1 and AT2 markers or in those clusters that were not clearly identifiable as AT2 or AT1.”

-now lines 634-649 read as follows: “Since our aim was to identify ADI cells, we also investigated the mean expression levels of all individual genes in the ADI cell marker list in each cluster (Supplementary Table 3) to support the module score analysis results. The bar plots with statistical analysis and the genes expression distribution within clusters are given in Supplementary Fig. 28-29 and Supplementary Table 4, 5, and 6. At 5 dpi, clusters 2 and 6 displayed significantly high mean expression levels for at least 8 ADI genes (cluster 2: 8/28 genes; cluster 6: 19/28 genes; Supplementary Fig. 28 and Supplementary Table 4 and 5). Cluster 5 did not show significantly high mean expression levels for any of the investigated ADI genes. These results partly confirmed the module score results and suggest that both AT1 and ADI cells belong to clusters 2 and 6.

At 14 dpi, clusters 0, 2, 3, 5, 7, and 9 displayed significantly high mean expression levels for at least 3 ADI genes (cluster 0: 6/28 genes; cluster 2: 10/28 genes; cluster 3: 3/28 genes; cluster 5: 9/28 genes; cluster 7: 19/28 genes; cluster 9: 9/28 genes; Supplementary Fig. 29 and Supplementary Table 4 and 6). These results confirmed the module score results and suggest that both AT1 and ADI cells belong to clusters 2, 5, and 6 as well as ADI cells belong to clusters 0, 3, and 9.”

6) ***“596 CoV-2-infected hamsters. At 5 dpi, ADI cells did not form a separate cluster, but were admixed with AT1 and AT2 cells. At 14 dpi, ADI cells were more numerous and clustered separately from AT1 and AT2 cells.” This statement cannot be included in this form: it is neither statistically backed nor, in many cases, easy to support even with further analyses. Note that UMAP does not necessarily preserve distances in the original space.***

We apologize to Reviewer 3 for the shortcoming. We just would like to underline the fact that we do not refer to distance but to separate clusters that have higher ADI module scores than others. In order to improve clarity we re-phrased the statement accordingly.

The following changes have been made:

RESULTS

-now lines 650-658 read as follows: “Taken together, transcriptome analysis identified clusters showing high module scores for AT1, AT2, ADI, club and ciliated cell marker gene sets. Additionally, we identified clusters with a significantly higher expression levels of ADI marker genes compared to other clusters. We interpret these differences in expression levels as absence or presence of the respective cell populations. Based on this, we assigned one or more presumable cell identities for each cluster (Fig. 8 A-B). We postulate that ADI cells showed an overlap of gene expression with AT1 cells, particularly at 5 dpi. At 14 dpi, a more distinct ADI cell population could be detected, which separated from the AT1 and AT2 clusters.

7) ***“ Given that proliferation is an effect in this system, it would be helpful to compute cell cycle scores and compare those against your cell identity scores to get an intuition for confounding sources of variation.”***

We thank Reviewer 3 for this suggestion. For this reason, we added the cell proliferation hallmark gene set G2M_CHCHECKPOINT to our analysis and compared the distribution of high scores to our cell identity scores.

The following changes have been made:

RESULTS

-now lines 693-696 read as follows: “Finally, we wanted to investigate which cell clusters show high scores for *G2M checkpoint* hallmark genes as a readout of cell proliferation. The highest scores for this gene set were observed in clusters with high AT2/club cell and AT1/ADI cell module scores at 14 dpi (Fig. 8 Q-R).”

8) - ***“629 In summary, the findings from the independent study confirmed that ADI cells are a feature of alveolar regeneration in hamsters on a transcriptome level,”***
As outlined above, I do not think that this is a direct conclusion from the presented analyses.

The authors understand Reviewer 3 concerns. However, after the additional analysis provided, based on Reviewer 3 valuable suggestions, we feel that these

results are supporting the presence of ADI cells in the hamster model for COVID-19. We also added a paragraph stating the limitations of our analysis.

- 9) ***“Reasoning based on scRNA-seq data: In general, this section is very descriptive. I would recommend the authors to re-work their line of reasoning here to specifically converge on a hypothesis in each section. As is now, the text mostly describes variation that exists in the data, which is not necessarily causally linked to their hypotheses.”***

We thank Reviewer 3 for this comment. We agree with Reviewer 3 point of view and for this reason all their suggestions have been thankfully taken into consideration in order to improve the scientific content of our work. As already outlined above in comment #1, this manuscript has been thoroughly revised by adding new analyses, increasing clarity, and stating clear hypotheses and aims when needed, in order to better guide the reader. We sincerely hope that the revised version of our manuscript will satisfy Reviewer 3 requests and will meet their approval.

- 10) ***“ Fig. 6 and 7 have very bad image quality in the merged pdf and are almost not readable, please look into that.”***

We apologize for this. We submitted a new version with a proper resolution.

A copy of the new Figures 7 and 8 have been attached to this document in comment #1. These figures will also be submitted separately, when allowed by the submission system.

- 11) ***“ 550 “First, we generated a Uniform Manifold Approximation and Projection (UMAP) clustering all cell populations detected in the datasets.” The UMAP does not cluster cells, it is an embedding. Please correct this phrasing and declare the type of clustering you are using here.”***

We apologize to Reviewer 3 for the shortcoming. The sentence has been revised accordingly. Uniform Manifold Approximation and Projection (UMAP) were

performed using default settings in Seurat package for clustering with functions FindNeighbors(pbmc, dims = 1:10) and FindClusters(pbmc, resolution = 0.5).

The following changes have been made:

RESULTS

-now lines 563-566 read as follows: “First, we generated a Uniform Manifold Approximation and Projection (UMAP) embedding using the Seurat package (functions: RunPCA, FindNeighbors, FindClusters⁴⁴⁻⁴⁹) for clustering to visualize and identify cell populations based on similar and known/presumed expression profiles”

REVIEWERS' COMMENTS

Reviewer #1 (Remarks to the Author):

Previous comments are addressed adequately with necessary details now added via links to challenge data, virus challenge information, etc.

The message of the manuscript remains the same, that hamsters may serve as a relevant model of lung damage and repair processes following SARS-COV-2 infection.

The data and the model are important.